# Apical anchorage and stabilization of subpellicular microtubules by apical polar ring ensures *Plasmodium* ookinete infection in mosquito

Pengge Qian[1,3], Xu Wang[1,3], Cuirong Guan[2,3], Xin Fang[1], Mengya Cai[1], Chuan-qi Zhong[1], Yong Cui[1], Yanbin Li[1], Luming Yao[1], Huiting Cui[1] ✉, Kai Jiang ●[2] ✉ & Jing Yuan ●[1] ✉

Morphogenesis of many protozoans depends on a polarized establishment of cortical cytoskeleton containing the subpellicular microtubules (SPMTs), which are apically nucleated and anchored by the apical polar ring (APR). In malaria parasite *Plasmodium*, APR emerges in the host-invading stages, including the ookinete for mosquito infection. So far, the fine structure and molecular components of APR as well as the underlying mechanism of APR-mediated apical positioning of SPMTs are largely unknown. Here, we resolve an unprecedented APR structure composed of a top ring plus approximate 60 radiating spines. We report an APR-localizing and SPMT-binding protein APR2. APR2 disruption impairs ookinete morphogenesis and gliding motility, leading to *Plasmodium* transmission failure in mosquitoes. The APR2-deficient ookinetes display defective apical anchorage of APR and SPMT due to the impaired integrity of APR. Using protein proximity labeling, we obtain a *Plasmodium* ookinete APR proteome and validate ten undescribed APR proteins. Among them, APRp2 and APRp4 directly interact with APR2 and also mediate the apical anchorage of SPMTs. This study sheds light on the molecular basis of APR in the organization of *Plasmodium* ookinete SPMTs.

Malaria remains a global infectious disease caused by unicellular apicomplexan protozoa of the genus *Plasmodium*, resulting in an estimated 627,000 deaths globally in 2020, a 12% increase in malaria death over 2019[1]. Transmission of a malaria parasite relies on its successful infection and development in the female *Anopheles* mosquito vector. Once entering the mosquito midgut after a blood meal, gametocytes are immediately activated to gametes that fertilize to form the zygotes. Within 12–24 h, the spherical zygotes undergo remarkable morphogenesis of "protrusion-elongation-maturation" to differentiate into crescent ookinetes[2–4]. Only mature ookinetes possessing gliding motility are capable of penetrating the mosquito midgut wall to colonize at the basal lumen where thousands of sporozoites develop within an oocyst for parasite transmission[5,6].

The *Plasmodium* ookinete, as well as two other invasive "zoite" stages (the salivary gland- and hepatocyte-invading sporozoite and the erythrocyte-invading merozoite), possesses a cortical pellicle unique to apicomplexan organisms[7,8]. From outside to inside, the pellicle consists of a parasite plasma membrane, a double-membrane organelle inner

[1]State Key Laboratory of Cellular Stress Biology, School of Life Sciences, Faculty of Medicine and Life Sciences, Xiamen University, Xiamen 361102 Fujian, China. [2]The State Key Laboratory Breeding Base of Basic Science of Stomatology and Key Laboratory of Oral Biomedicine Ministry of Education, School & Hospital of Stomatology, Frontier Science Center for Immunology and Metabolism, Medical Research Institute, Wuhan University, Wuhan, China. [3]These authors contributed equally: Pengge Qian, Xu Wang, Cuirong Guan. ✉e-mail: cuihuiting@xmu.edu.cn; jiangkai@whu.edu.cn; yuanjing@xmu.edu.cn

membrane complex (IMC), and a layer of apically radiating subpellicular microtubules (SPMTs), both of which closely associate with each other and span along the periphery of the parasite[7,8]. Variable numbers of SPMTs are assembled in different zoite stages, around 60 SPMTs in ookinetes[4,8–12], 16 SPMTs in sporozoites[13], and 1–4 SPMTs in merozoites[14–16]. Elucidation of the fine structure of SPMTs array in the *Plasmodium* has long been hampered due to the limited resolution for this compact structure via conventional electron or fluorescence microscopy. Recently, Bertiaux *et.al* have applied elegant ultrastructure expansion microscopy (U-ExM) and observed a remarkable cell-size dome-like arrangement of SPMT array at an unprecedented resolution in the ookinetes[14]. In *Plasmodium*, the SPMT arrays play at least two essential roles. One is as a scaffold supporting parasite morphogenesis[4,17], maintaining the unusual cell shapes[13] and providing parasite rigidity during gliding and invasion[18]. The other is as a platform for docking the apical secretory organelles[17], whose protein secretion is through a putative apical gateway for parasite gliding and invasion. While the SPMT cytoskeleton is fundamental for parasite morphogenesis, gliding, and invasion, the mechanisms for its apical biogenesis, pellicle anchorage, and geometry regulation remain largely unknown.

Besides SPMTs, the invasive zoites of apicomplexan parasites also possess a highly conserved and specialized structure called the apical polar ring (APR) at the cell apical cortex. In the transmission electron micrograph of the *Plasmodium* ookinetes, APR has been described as an electron-lucent region beneath an electron-dense layer which is presumably the apical end of IMC[11,19,20]. APR resembles a cap-like structure with the minus-ends of all SPMTs embedded[20,21]. Since all the SPMTs emanate from APR, it has been widely accepted that APR functions as a microtubule-organizing center (MTOC) for nucleating and anchoring SPMTs at the *Plasmodium* zoites[8,22–24]. It is reasonable to speculate that APR proteins regulate the nucleation, assembly, anchorage, and geometry of SPMT, but none has yet been identified and functionally verified. So far, only two proteins, ARA1 (apical ring associated protein 1) and APR2 have been reported to display an APR-like localization by immunoelectron or fluorescence microscopy in the ookinetes of *P. berghei*[25,26]. The orthologues of APR2 are encoded in many apicomplexan parasites, while ARA1 is unique to the *Plasmodium* parasites. However, the precise localization of ARA1 and APR2 at the APR and their functions in the parasite remain elusive.

Here we study the apicomplexan-conserved APR2 (PY17X_1339500) in the rodent malaria parasite *P. yoelii* and perform in-depth biochemical and phenotypical analyses. We demonstrate that APR2 is an APR-residing MT-binding protein that plays an essential role in parasite transmission in mosquitoes. APR2, with its APR-residing partners APRp2 and APRp4 identified by proximity labeling and yeast two-hybrid, coordinately regulate the structural integrity of the APR to ensure apical anchorage and integrity of SPMTs, which serves a critical role in the ookinete development, shape, and motility.

## Results

### APR2 is essential for mosquito transmission of the *P. yoelii* parasite

To elucidate the function of APR2 in the life cycle of parasite, we disrupted the *apr2* gene (PY17X_1339500) in *P. yoelii* 17XNL strain (wildtype) using the CRISPR-Cas9 method and obtained 2 mutant strains Δ*apr2* and Δ*apr2n*. The Δ*apr2* had the whole *apr2* coding sequence (4.1 kb) deleted, while the Δ*apr2n* had a 601 bp deletion in the 5′ coding region, causing a frameshift for the remaining coding sequence (Fig. 1a). Both Δ*apr2* and Δ*apr2n* exhibited comparable levels as wildtype (WT) in asexual blood stage proliferation and gametocyte formation in mice (Fig. 1b, c). To evaluate the role of APR2 in parasite development in mosquitoes, *Anopheles stephensi* mosquitoes were fed on the parasite-infected mice. Neither Δ*apr2* nor Δ*apr2n* produced midgut oocysts on day 7 post-infection (pi) (Fig. 1d), indicating parasite transmission failure in the mosquito. Consistent with this finding, no

sporozoites in the salivary glands on day 14 pi (Fig. 1e) and no transmission from mosquitoes to mice were observed (Fig. 1f). To confirm the transmission defects were caused by APR2 deletion, we generated the complemented strain *comp* by introducing the deleted 601 bp sequence fused with a sextuple hemagglutinin (6HA) coding sequence back into the *apr2* locus of the Δ*apr2n* mutant (Fig. 1a). Expression of 6HA-tagged APR2 was detected in the *comp* strain using immunoblot (Fig. 1g). As expected, the *comp* strain restored the oocyst formation in mosquitoes (Fig. 1h).

### APR2 regulates ookinete development, shape, and gliding motility

APR2 null parasites developed gametocytes but no oocyst, indicating defects in one or more steps between gametocyte and oocyst stages. We performed in-depth experiments to delineate the step(s) affected by APR2 deficiency. The Δ*apr2* showed normal levels of both male and female gamete formation and fertilization in vitro compared with the WT parasite (Supplementary Fig. 1a–c). Measurement of DNA content after Hoechst 33342 staining indicated that female gametes of Δ*apr2* could fertilize and further develop from diploid to tetraploid (Supplementary Fig. 1d, e). However, the in vitro assay for zygote to ookinete differentiation revealed that Δ*apr2* had a dramatic decrease in mature ookinete formation (62% in WT; 20% in Δ*apr2*), while the complemented parasite restored normal levels (Fig. 1i). We also isolated parasites from infected mosquito midguts and detected similar defects of Δ*apr2* in vivo (Fig. 1j). The ookinetes from WT and *comp* parasites were crescent-shaped; however, this shape was lost in all of the Δ*apr2* and Δ*apr2n* ookinetes with less cell bending (Fig. 1k), suggesting an alteration in the ookinete cytoskeleton in the absence of APR2.

Gliding motility is a prerequisite for midgut traversal of ookinetes. We assessed the gliding capability of the Δ*apr2* mature-looking ookinetes using an in vitro Matrigel-based assay. Notably, the Δ*apr2* and Δ*apr2n* ookinetes displayed a significantly reduced gliding speed compared to WT and *comp* (WT: $6.5 \pm 2.5\,\mu\text{m/min}$, $n = 38$; Δ*apr2*: $1.0 \pm 1.3\,\mu\text{m/min}$, $n = 39$; Δ*apr2n*: $1.3 \pm 1.4\,\mu\text{m/min}$, $n = 25$; *comp*: $6.8 \pm 3.0\,\mu\text{m/min}$, $n = 28$) (Fig. 1l). As a control, the ookinetes lacking the gliding-essential gene *ctrp* (23, 24), completely lost motility. Since ookinete development, shape and gliding were severely impaired, we speculated that the *apr2*-deficient parasites may fail to traverse the mosquito midgut. To test this, the midguts from infected mosquitoes were dissected at 24 h pi and visualized after staining with an antibody against P28 (plasma membrane protein in ookinetes and early oocysts). The numbers of P28-positive (P28+) parasites were significantly reduced in Δ*apr2*-infected midguts compared with WT-infected midguts (parasites per mosquito: $90 \pm 58$ in WT, $n = 32$; $3 \pm 6$ in Δ*apr2*, $n = 40$) (Fig. 1m). These results demonstrate that APR2 regulates ookinete development, shape, and gliding for parasite establishing an infection in the mosquito midgut (Fig. 1n).

### APR2 localizes at APR throughout zygote to ookinete development

To investigate the expression and localization of APR2 during ookinete development, we tagged the APR2 with a 6HA at either the N- or C-terminus in the 17XNL parasite using the CRISPR-Cas9 method, generating *6HA::apr2* and *apr2::6HA* clones (Supplementary Fig. 2a). Immunofluorescence assay (IFA) of cultured ookinetes showed that APR2 was localized at the apical end during the whole development process (Fig. 2a). APR2 expression was detected in other parasite stages, including asexual blood stage, gametocyte, midgut oocyst, and salivary gland sporozoite (Supplementary Fig. 2a–c). Apical localization of APR2 was also observed in merozoites and sporozoites (Supplementary Fig. 2c), suggesting APR2 as an APR protein in all three zoites. We generated another two independent clones *gfp::apr2* and *apr2::gfp* with endogenous APR2

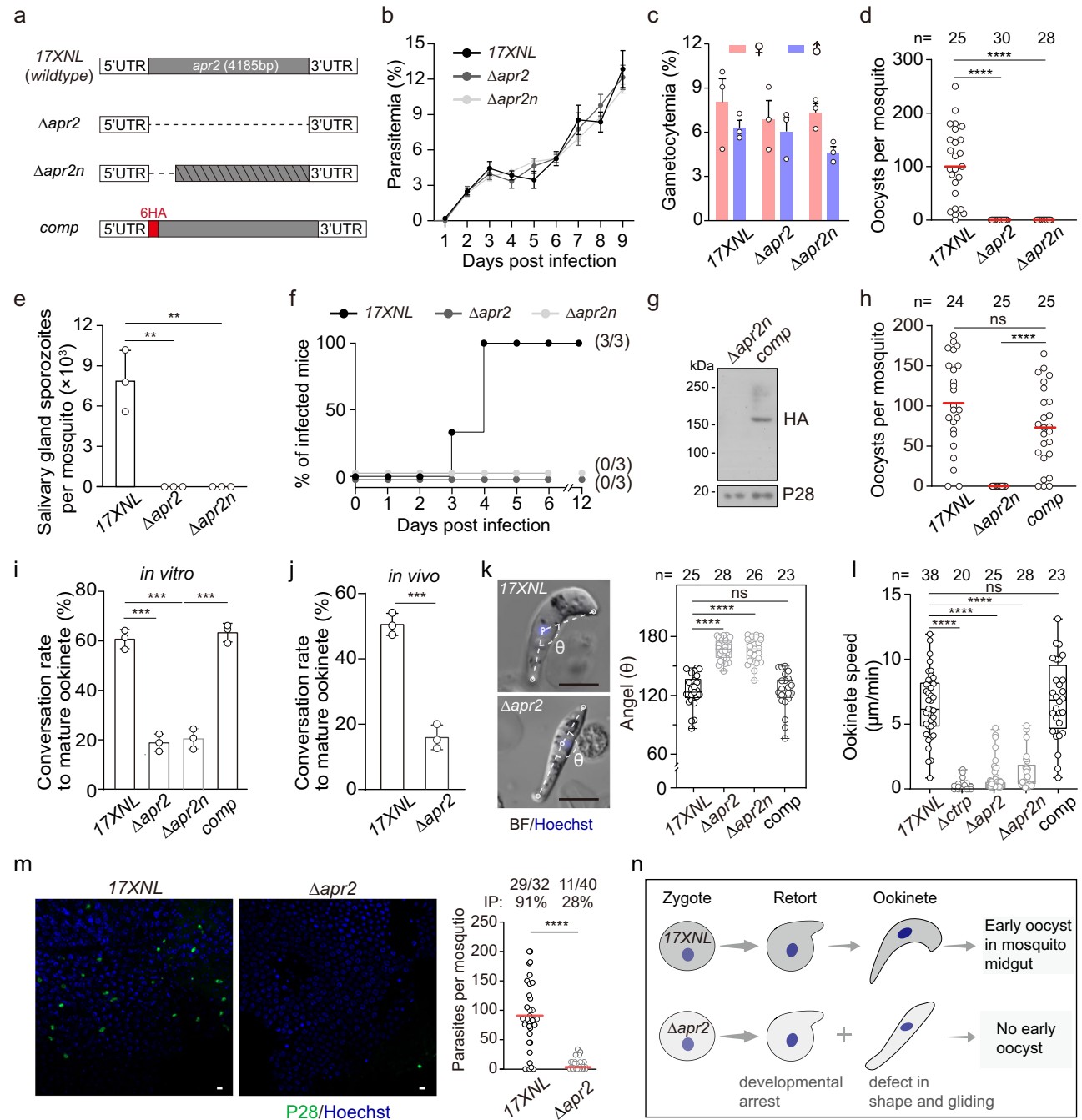

tagged with GFP at the N- and C-terminus, respectively, and observed a similar apical localization pattern of APR2 throughout ookinete development (Fig. 2b). To compare the APR2 localization to proteins localized at the apical tubulin ring (ATR), a structure at the apical extremity of ookinetes[14,25], we genetically tagged the ATR-localized proteins MyosinB and SAS6L respectively with a quadruple Myc epitope (4Myc) from the *apr2::6HA* parasite[27,28]. Under the confocal microscope, MyosinB and SAS6L marked the cellular localization of ATR, and APR2 was only detected at APR but not at ATR (Fig. 2c). These results fit with the distinct positions of APR and ATR in the electron microscopy images of ookinetes (Supplementary Fig. 3a)[14,29]. Additionally, the pellicle proteins P28, GAP45 (IMC), and both α-and β-Tubulin showed overlapping with APR2 only at the apical periphery of ookinete (Fig. 2c), suggesting that APR2 was strictly localized at APR but not at the outmost PM or IMC. The microneme proteins chitinase and CTRP did not show co-

localization with APR2 (Fig. 2c). Moreover, immunoelectron microscopy (immuno-EM) analysis of the *apr2::6HA* ookinetes showed that the colloidal gold-labeled APR2 were detected at APR in both longitudinal and transverse section images (Fig. 2d).

To visualize APR2 localization at APR in more detail, the *apr2::6HA* parasites from 3- and 12-h in vitro ookinete culture were stained with protein NHS-ester dye (NHS)[14] and anti-HA antibody and visualized by the ultrastructure expansion microscopy (U-ExM). In early ookinetes with an initial protrusion, an apical thin APR2-staining ring emerged, overlapping with the NHS dense-stained area (Fig. 2e). In mature ookinetes, many short HA-labeled spines grew from the apical ring and radiated in a posterior direction, overlapping with the NHS dense-stained fiber-like structure (Fig. 2e). The apical ring plus spines pattern of APR2 under high resolution in ookinete was in accordance with the cap-like signal of APR2 under low resolution (Fig. 2a, b).

**Fig. 1 | APR2 is essential for mosquito transmission of the *P. yoelii* parasite.**
**a** Diagram showing genetic deletion and complementation of the *apr2* gene using the CRISPR-Cas9 method. Full length and N-terminus of the *apr2* gene were deleted respectively in the 17XNL strain, generating two mutant clones Δ*apr2* and Δ*apr2n*. The Δ*apr2n* mutant was complemented by introducing the deleted sequence of *apr2* gene fusing with a N-terminal sextuple HA epitope (6HA), generating the complementation clone *comp*. UTR, untranslated region. **b** Parasite intraery-throcytic proliferation in mice. Values are means ± SEM ($n = 3$ biological replicates).**c** Gametocyte formation in mice. Values are means ± SEM ($n = 3$ biological replicates). **d** Midgut oocysts formation in mosquito 7 days post infection (dpi). *n* is the number of mosquitoes dissected. Representative results from two independent experiments. Red horizontal lines show mean value of oocyst numbers. From left to right: ****$P = 3e-11$ and ****$P = 2e-10$, by two-sided Mann–Whitney test. **e** Salivary gland sporozoite count in mosquito 14 dpi. At least 20 infected mosquitoes were counted in each group per replicate. Values are means ± SEM ($n = 3$ biological replicates). From left to right: **$P = 4e-03$ and **$P = 4e-03$, by two-sided *t* test. **f** Infectivity of sporozoites from mosquito to mice via natural biting. Infected mice were determined by emergence of parasites monitored by blood thin smear. x/y in the bracket indicated the number of infected mice and total naïve mice used. **g** Immunoblot of the HA-tagged APR2 in the ookinetes of the Δ*apr2n* and *comp* parasites ($1.0 \times 10^6$ ookinetes were lysed for each sample). P28 is a loading control. Two independently performed experiments with similar results. **h** Midgut oocysts in mosquitoes infected with the *comp* strain 7 dpi. n is the number of mosquitoes dissected. Red horizontal lines show the mean value of oocyst numbers. From left to right: ns $P = 0.05$ and ****$P = 6e-12$, by two-sided Mann–Whitney test. **i** Mature ookinete formation in vitro. Values are means ± SEM ($n = 3$ biological

replicates). From left to right: ***$P = 2e-04$, ***$P = 2e-04$, and ***$P = 2e-04$, by two-sided *t* test. **j** Mature ookinete formation in the mosquito. Values are means ± SEM ($n = 3$ biological replicates). ***$P = 3e-04$, two-sided *t* test. **k** Ookinete cell shape. Three points were selected in each ookinete: apical end point, nucleus center point, and basal end point. The nucleus center point was set as vertex of the angel (θ) from two lines of nucleus-apical and nucleus-basal. n is the number of ookinetes analyzed. Scale bars: 5 μm. Quantification of θ in right panel. Representative results from two independent experiments. Boxes show medians with interquartile ranges, whiskers: min to max show all points. From left to right: ****$P = 3e-15$, ****$P = 2e-13$, ns $P = 0.99$, respectively, by two-sided Mann–Whitney test. **l** Ookinete gliding motility using the in vitro Matrigel-based assay. Shown were one representative result from two independent experiments. *n* is the number of ookinetes analyzed. Boxes show medians with interquartile ranges, whiskers: min to max show all points. From left to right: ****$P = 4e-15$, ****$P = 3e-19$, ****$P = 2e-13$, and ns $P = 0.58$, by two-sided Mann–Whitney test. **m** Immunofluorescence analysis (IFA) of P28 in ookinete and early oocyst at mosquito midguts infected with 17XNL and Δ*apr2* 24 h pi. P28 is a plasma membrane protein of ookinete and early oocyst. Right panel showed the quantification of parasites per midgut. The numbers on the top are the count of midguts containing parasite/the count of midguts measured; IP, infection prevalence; red horizontal lines show the mean value of parasite numbers. ****$P = 5e-14$, two-sided Mann–Whitney test. Two independently performed experiments with similar results. Scale bars: 5 μm. **n** Cartoon illustrating APR2 deficiency in ookinete development, shape, and gliding during mosquito transmission of parasite. Approximately two thirds of APR2-null ookinetes arrest at the early stages and the rest develop to ookinetes with less cell bending and defective gliding.

## APR2 associates with apical SPMTs

Since APR2 localizes at APR, we asked whether APR2 associates with apical SPMTs. First, co-immunostaining of APR2 (HA epitope) and SPMT (alpha- and beta-Tubulin) in the *apr2::6HA* parasites detected apical co-localization of APR2 with SPMTs throughout ookinete development (Fig. 3a). Second, proximity ligation assay (PLA) revealed PLA signals at apical of the ookinetes in both *apr2::6HA* and *6HA::apr2* parasites but not in WT parasite when both anti-Tubulin and anti-HA antibodies were present (Fig. 3b), indicative of close proximity between APR2 and apical SPMTs. Third, co-immunoprecipitation (Co-IP) with anti-HA antibody indicated that APR2 binds to SPMTs in the *apr2::6HA* ookinete lysates (Fig. 3c). Fourth, we analyzed the association of APR2 in the ookinete ghost after extraction with the ionic detergent sodium deoxycholate (SDC) (Fig. 3d), which was used for isolating SPMT cytoskeleton in the *T. gondii*[30]. After SDC treatment, the *apr2::6HA* ookinete SPMT cytoskeleton remained intact, recognizable as bright fluorescent signals of Tubulin (Fig. 3e), whereas the pellicle membranes (PM and IMC) proteins were largely depleted (Supplementary Fig. 3b). On the contrary, APR2 remained at apical in SDC-treated *apr2::6HA* ookinetes (Fig. 3e), indicating close association with ookinete ghost. Lastly, we used the U-ExM to visualize APR2 localization with respect to SPMTs after staining with anti-Tubulin and anti-HA antibodies. In early ookinetes, several SPMTs radiated from a small APR2 ring, likely the early structure of APR (Fig. 3f, upper panel). In mature ookinetes, both the SPMTs array and the ATR were established (Fig. 3f, lower panel), in accordance with those observed in the *P. berghei* ookinetes[14]. Notably, APR2 was apically co-localized with SPMTs but not with ATR (Fig. 3f, lower panel), consistent with the APR-SPMT connection in the electron microscopy image of ookinetes (Supplementary Fig. 3a). These results showed that APR2 closely associates with apical SPMTs.

Considering APR2 tightly associating with apical SPMTs, we speculate that APR2 may be localized at APR in a fixed position with less turnover. Using fluorescence recovery after photobleaching (FRAP), we found that the fluorescent signal of APR2-GFP at APR in the *apr2::gfp* ookinetes after photobleaching could not be recovered even after minutes (Supplementary Fig. 3c, d). While in the control parasite line *gfp* with transgenic GFP expressed in the cytosol[31], the GFP signal at the apical areas recovered within seconds after photobleaching

(Supplementary Fig. 3c, d). These FRAP results confirmed that APR2 is localized in a fixed position at APR, likely in the contact region between APR and SPMTs.

## APR2 N-terminal region shows MT-binding and -stabilization properties

APR2 association with SPMTs prompted us to speculate that APR2 itself is an MT-binding protein. To prove this, we expressed GFP-tagged APR2 in human cells and tested whether it could co-localize with MT. If APR2 can associate with MT in a heterologous system devoid of other parasite proteins, then its binding to MT is likely in a direct manner. Indeed, GFP-tagged APR2 (APR2-GFP) co-localized with the MT fibers in the transfected MRC5 cell line (Fig. 4a). APR2 is a protein of 1394-amino acid residues with no recognizable domain or no significant homology to any other proteins. Next we sought to identify the APR2 sub-region(s) involved in MT binding. Three partly overlapping unstructured fragments (APR2-N: 1–600 aa, APR2-M: 501–1100 aa, and APR2-C: 1001–1394 aa) were fused with GFP at the C-terminus and transiently expressed in the MRC5 cells (Fig. 4a). Under the same conditions, only GFP-tagged APR2-N displayed a co-localization with MTs, while APR2-M and APR2-C did not (Fig. 4a). A similar MT-binding activity of APR2-N was observed in the HEK293T cell line (Fig. 4b). Co-immunoprecipitation assay also revealed that GFP-tagged APR2-N, but not APR2-M or APR2-C, bound to MTs in the HEK293T cells (Fig. 4c). To further validate the MT-binding activity of APR2 sub-regions, we purified the GFP-tagged APR2-N, APR2-M, and APR2-C from the HEK293T cells and incubated them with the poly-merized MTs in vitro. As expected, GFP-tagged APR2-N bound directly to both MT seeds and lattices in vitro, while APR2-M and APR2-C did not (Fig. 4d). Therefore, APR2 is an MT-binding protein and this property is contributed by the APR2-N fragment.

We further mapped the region within APR2-N for MT-binding (Supplementary Fig. 4a). Deletion analysis of APR2-N showed the N-terminal part of 200 aa likely a minimum region for MT-localization in mammalian cells (Supplementary Fig. 4b) and for in vitro MT binding (Supplementary Fig. 4c). Consistently, all the APR2-N trun-cated fragments harboring MT-binding activity were sufficient to tar-get to the APR when expressed episomally in the WT ookinetes (Supplementary Fig. 4d).

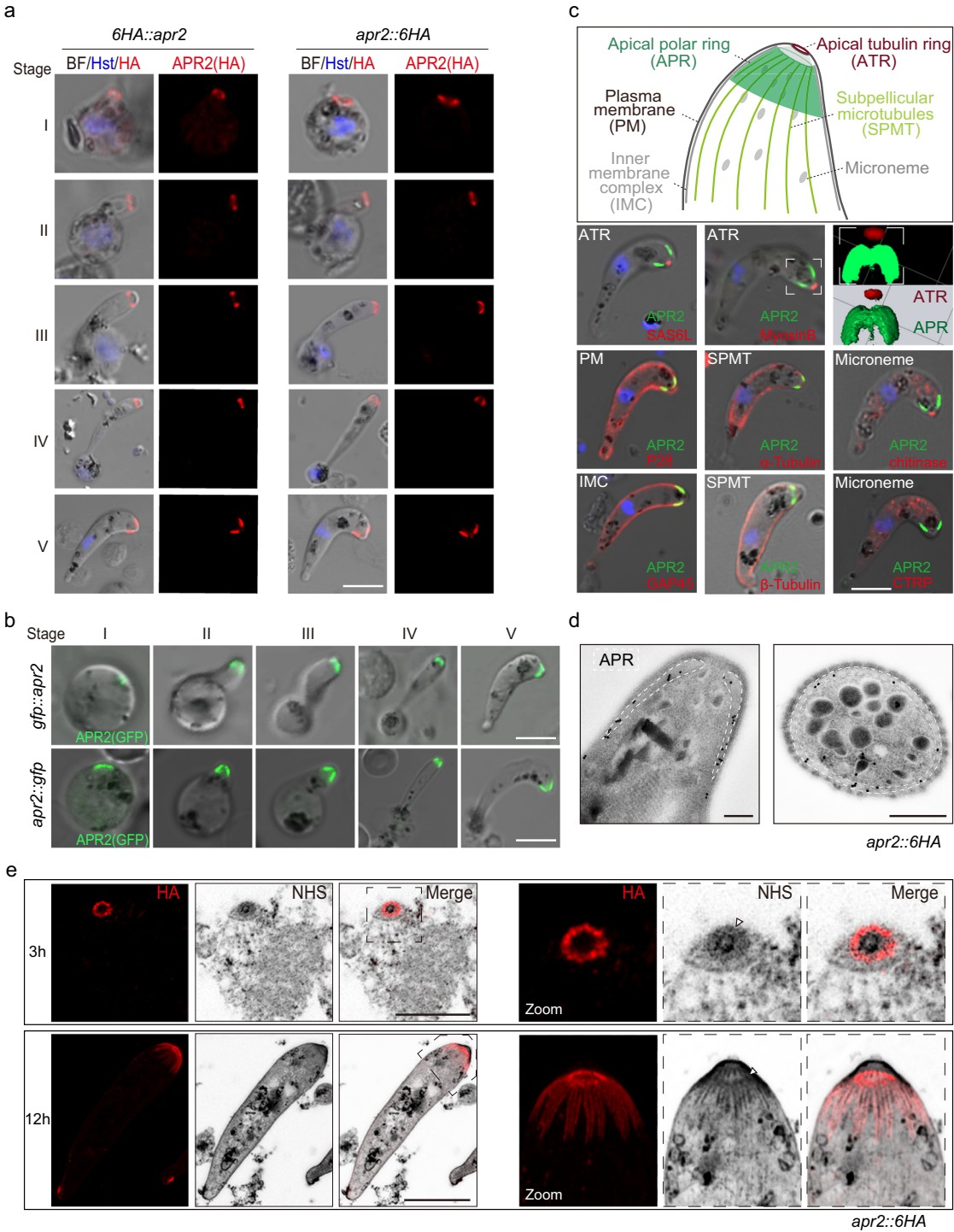

**Fig. 2 | APR2 localizes at APR throughout zygote to ookinete development.**
**a** IFA of APR2 expression during zygote (stage I) to ookinete (stage V) development of two HA-tagged strains *6HA::apr2* (left panel) and *apr2::6HA* (right panel). Three independently performed experiments with similar results. Scale bars: 5 μm.
**b** Fluorescence detection of APR2 in living parasites during zygote to ookinete development of two GFP-tagged strains *gfp::apr2* (top panel) and *apr2::gfp* (bottom panel). Two independently performed experiments with similar results. Scale bars: 5 μm. **c** Co-localization analysis by IFA for APR2 with proteins of known cellular localizations in ookinetes. P28 (plasma membrane, PM), GAP45 (inner membrane complex, IMC), α- and β-Tubulin (subpellicular microtubule, SPMT), MyosinB and SAS6L (apical tubulin ring, ATR), chitinase and CTRP (microneme). Top panel shows a diagram of apical structure of ookinete. APR2 is tagged with a 6HA. SAS6L

and MyosinB are tagged with a 4Myc. P28, GAP45, α- and β-Tubulin, chitinase, and CTRP are detected using the antibody or antiserum. Two independently performed experiments with similar results. Scale bars: 5 μm. **d** Immuno-EM images of the *apr2::6HA* ookinete. Gold particles coupled with the anti-HA antibody are located at APR (white dashed lined area) in the longitudinal view (left) and cross view (right). Two independently performed experiments with similar results. Scale bars: 0.5 μm. **e** Ultrastructure expansion microscopy (U-ExM) of APR2 in the *apr2::6HA* ookinetes from 3- and 12-h in vitro culture. The parasites were co-stained with the anti-HA antibody and the protein NHS-ester dye. Dashed-lined area containing NHS-ester dense region (triangle) were zoomed in and shown in the right panels. Three independently performed experiments with similar results. Scale bars: 5 μm.

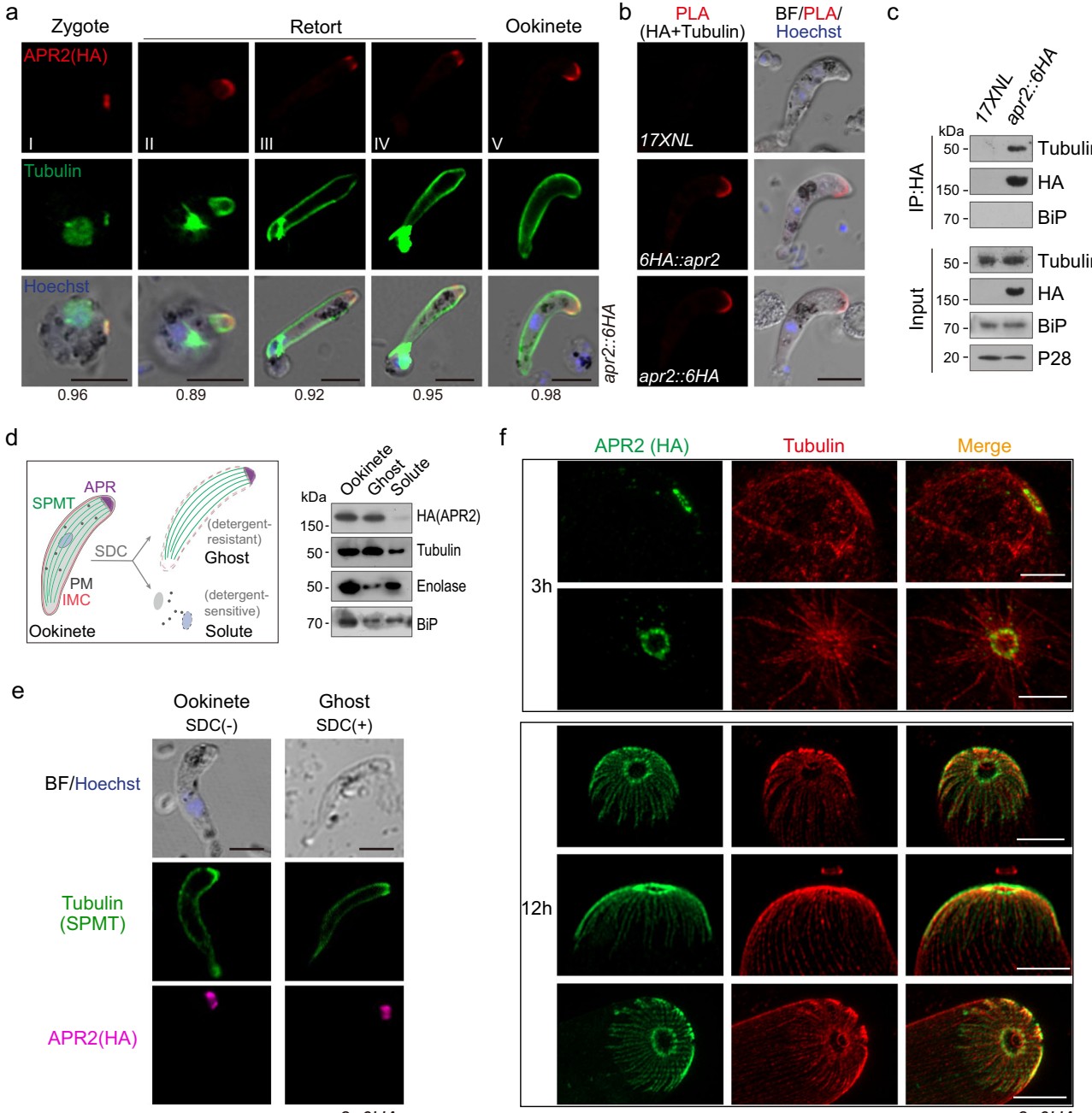

**Fig. 3 | APR2 associates with apical subpellicular MTs (SPMTs). a** IFA of APR2 (HA, red) and SPMT (α- and β-Tubulin, green) from zygote (stage I) to ookinete (stage V) development of the *apr2::6HA* parasite. The values in the bottom are the Pearson correlation coefficient for the co-localization between APR2 and apical SPMTs. Three independently performed experiments with similar results. Scale bars: 5 μm. **b** Proximity ligation assay (PLA) detecting protein interaction between APR2 and Tubulin in the *6HA::apr2* and *apr2::6HA* ookinetes. Two independently performed experiments with similar results. Scale bars: 5 μm. **c** SPMTs co-immunoprecipitated with HA-tagged endogenous APR2 in *apr2::6HA* ookinetes. Co-immunoprecipitation was conducted using anti-HA antibody. P28 and BiP as the loading control. Representative result from two biological replicates. **d** Isolation of ookinete SPMT cytoskeleton (ookinete ghost). Left panel indicated the work-flow.

$5.0 \times 10^6$ ookinetes are treated with the ionic detergent sodium deoxycholate (SDC), and the ghost fraction containing cytoskeleton (detergent-resistant) and solute fraction (detergent-sensitive) were collected respectively. Immunoblot of the APR2 and Tubulin in the ookinete lysate, ghost fraction, and solute fraction. ER protein BiP and cytosolic protein enolase are used as loading control. Two independently performed experiments with similar results. **e** IFA of APR2 (HA, magenta) and SPMT (α- and β-Tubulin, green) in the *apr2::6HA* ookinetes and ookinete ghosts. Three independently performed experiments with similar results. Scale bars: 5 μm. **f** U-ExM of APR2 (HA, green) and SPMT (α- and β-Tubulin, red) in the *apr2::6HA* ookinetes from 3- and 12-hour in vitro culture. Two independently performed experiments with similar results. Scale bars: 1 μm.

In the in vitro MT-binding assay, we noticed the altered dynamics of MT growth behaviors, including decreased depolymerization and catastrophe of MT and increased rescue of MT, when inoculating with APR2-N compared with control and counterparts APR2-M and APR2-C (Fig. 4e). These results suggested that APR2-N confers an MT-stabilizing activity. Therefore, we performed a titration in APR2-N concentration from 10 nM to 50 nM in the in vitro MT-binding assays and confirmed its MT-stabilizing activity increases in a dose-dependent manner (Fig. 4f, g). Interestingly, APR2-N also preferentially binds to GMPCPP-stabilized seeds in vitro at low

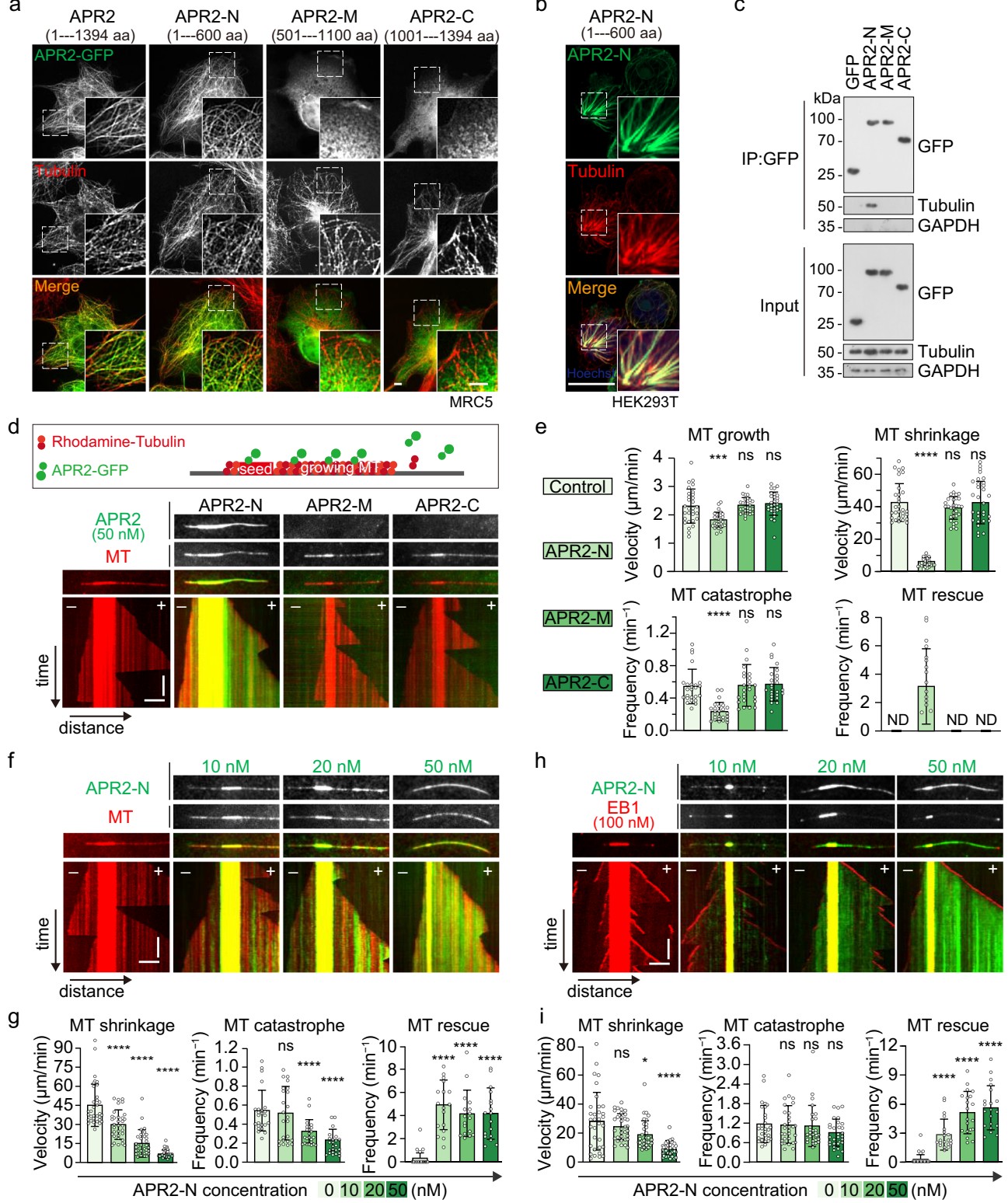

concentrations (Fig. 4f). Moreover, we evaluated the MT-stabilizing activity of APR2-N in the presence of human EB1 (100 nM), which has been shown to facilitate MT catastrophe in vitro[32,33]. In the presence of EB1, APR2-N still executed a protection effect on MT stabilization in a dose-dependent manner (Fig. 4h, i). These results showed that APR2-N is able to stabilize MTs in vitro.

Strikingly, we found that GFP-tagged APR2 and APR2-N preferentially accumulated at the curved MTs in certain transfected MRC5 cells (Supplementary Fig. 5a). The MT geometry preference

of APR2-N is reminiscent of that of WDR62, which also preferentially associates with the curved MTs in mammalian cells[34]. Therefore, we examined the relative distributions of APR2-N and WDR62 tracked by tagged-GFP and mCherry respectively in co-transfected MRC5 cells. As expected, APR2-N and WDR62 displayed a co-localization pattern at the curved MTs (Supplementary Fig. 5b). We next investigated the MT-binding preference of APR2 in vitro by producing curved MTs in vitro (see Material and Method for details). GFP-tagged APR2-N bound along the MTs, with stronger accumulation at

**Fig. 4 | APR2 N-terminal region shows MT-binding and MT-stabilizing properties. a** IFA of GFP-tagged APR2 (green) and MTs (red) in the mammalian cell line MRC5. Full-length APR2 and three overlapping fragments (APR2-N: 1–600 aa, APR2-M: 501–1100 aa, and APR2-C: 1001–1394 aa) were fused with GFP at the C-terminal and transiently expressed in the MRC5 cells. Three independently performed experiments with similar results. Scale bars: 5 μm. **b** IFA of the GFP-tagged APR2-N (green) and MTs (red) in the mammalian cell line HEK293T. Three independently performed experiments with similar results. Scale bars: 5 μm. **c** Co-immunoprecipitation of GFP-tagged APR2 fragments (APR2-N, APR2-M, and APR2-C) and MTs in HEK293T cell. Immunoprecipitations were performed using antibody against GFP. GFP serves as a negative control. GAPDH is a loading control. Two independently performed experiments with similar results. **d** In vitro MT binding of proteins detected by total internal reflection fluorescence (TIRF) microscopy. A diagram in upper panel shows the GMPCPP-stabilized MT (red) as nucleation seed for MT growth in the presence of tested proteins (green). Representative images and kymographs showed MT-binding property of APR2-N, but not APR2-M and APR2-C. 50 nM of each APR2 fragment was used. Horizontal scale bars: 2 μm; vertical scale bars: 1 min. Two independently performed experiments with similar results. **e** Quantification of MT dynamic events in **d**, including MT growth, shrinkage, catastrophe, and rescue in the presence of APR2-N, APR2-M, and APR2-C. APR2-N caused a minor decrease in MT growth, but had a marked effect in MT shrinkage and catastrophe. Values are means ± SEM ($n$ = 3 biological replicates).

From left to right for MT growth: ***$P$ = 2e−04, ns $P$ = 0.91, and ns $P$ = 0.56. From left to right for MT shrinkage: ****$P$ = 6e−24, ns $P$ = 0.13, and ns $P$ = 0.81. From left to right for MT catastrophe: ****$P$ = 6e−08, ns $P$ = 0.84, and ns $P$ = 0.65. Two-sided $t$ test applied. ND: not detected. **f** Representative images and kymographs showing MT-stabilizing property of APR2-N in a dosage-dependent manner. Horizontal scale bars: 2 μm; vertical scale bars: 1 min. Two independently performed experiments with similar results. **g** Quantification of MT dynamic events in **f**, including MT shrinkage, catastrophe, and rescue in the presence of a titration of APR2-N. Values are means ± SEM ($n$ = 3 biological replicates). From left to right for MT shrinkage: ****$P$ = 1e−19, ****$P$ = 4e−13, and ****$P$ = 3e−05. From left to right for MT catastrophe: ns $P$ = 0.55, ****$P$ = 5e−05, and ****$P$ = 6e−08. From left to right for MT rescue: ****$P$ = 4e−10, ****$P$ = 1e−08, and ****$P$ = 3e−08. Two-sided $t$ test applied. **h** Representative images and kymographs showing MT-stabilizing property of APR2-N in the presence of 100 nM EB1. EB1 facilitates MT catastrophe in vitro. Horizontal scale bars: 2 μm; vertical scale bars: 1 min. Two independently performed experiments with similar results. **i** Quantification of MT dynamic events in **h**, including MT shrinkage, catastrophe, and rescue, upon addition of a titration of APR2-N and 100 nM EB1. Values are means ± SEM ($n$ = 3 biological replicates). From left to right for MT shrinkage: ns $P$ = 0.28, *$P$ = 0.02, and ****$P$ = 1e−06. From left to right for MT catastrophe: ns $P$ = 0.85, ns $P$ = 0.61, and ns $P$ = 0.05. From left to right for MT rescue: ****$P$ = 2e−07, ****$P$ = 5e−11, and ****$P$ = 9e−12. Two-sided $t$ test applied.

the curved regions than at the straight parts (Supplementary Fig. 5c). These results suggested that APR2 has a preferential binding to the curved MTs.

## APR2 C-terminal region could localize at APR

Whether APR2-N (1–600 aa) with MT-binding activity is sufficient to target APR2 to the APR remains unknown. To test this, we deleted the coding region of 2–600 aa of endogenous APR2 in the *apr2::gfp* parasite, generating a modified clone *APR2-ΔN* (Fig. 5a). Surprisingly, the C-terminal part (601–1394 aa) of APR2 in the *APR2-ΔN* ookinetes still showed APR localization, although its fluorescent signal at APR was greatly decreased compared with that of the full-length APR2 in the *apr2::gfp* ookinetes (Fig. 5b, c). Therefore, besides APR2-N, other sub-region(s) of APR2 could associate with the APR. To identify such sub-region(s), APR2-M and APR2-C were fused with a 6HA tag at the C-terminus and episomally expressed in the WT parasites (Fig. 5d). APR2-C was localized at APR, while APR2-M was in the cytosol (Fig. 5e). We also examined the localization of APR2-M and APR2-C in the absence of endogenous APR2. In the *Δapr2* ookinetes, episomally expressed APR2-M and APR2-C displayed the localization pattern similarly as in WT ookinetes (Fig. 5e). These results indicated that APR2-C also has an APR-localizing property, which is independent of the endogenous APR2. Notably, APR2-C at APR showed a smaller fluorescence signal area in *Δapr2* compared to WT ookinetes (Fig. 5e, f), similar to the phenomenon observed in Fig. 5b, c. A significant reduction in protein occupancy at APR suggested an altered structure of APR in APR2-deficient ookinetes.

To further confirm the independent APR-targeting ability of APR2-N and APR2-C, we inserted a T2A "self-cleaving" peptide into the endogenous APR2 between amino acid site 845/846 in the 17XNL parasite, generating the *APR2-T2A* parasite line (Fig. 5g). In this editing, the N-terminal part (1–845 aa covering APR2-N) and the C-terminal part (846–1394 aa covering APR2-C) were fused with HA and Myc epitope, respectively. Immunoblot detected two bands with expected molecular mass (Fig. 5h), confirming the separation of N- and C-terminal parts of APR2 by T2A. Both the N and C-terminal parts displayed apical localization in early and mature ookinetes of the *APR2-T2A* parasite (Fig. 5i). Together, these results demonstrated that APR2-C alone is capable of targeting the APR (Fig. 5j). Considering no detectable MT-binding activity of APR2-C, we reasoned that APR2-C targets to the APR by interacting with other APR-localizing proteins.

## APR2 is required for integrity and apical anchorage of APR-SPMT

APR localization and SPMT association of APR2 as well as its implication in ookinete development, shape, and gliding suggest roles of this protein in regulating APR, SPMT, or both. Aberrant apical cytoskeleton caused by APR2 deficiency was also reflected by distinct apical morphology between WT and *Δapr2* ookinetes under the scanning electron microscopy (SEM). While WT mature ookinetes (100%, $n$ = 30) had apex surface with continuous and smooth radian (Fig. 6a, upper panel), approximately 90% of mature-looking *Δapr2* ookinetes ($n$ = 45) developed a collapsed apex (Fig. 6a, lower panel). Examination of the apical pellicle, APR, and SPMTs by transmission electron microscopy (TEM) revealed intact IMC underlying PM in the *Δapr2* ookinetes (Fig. 6b), which was also confirmed by GAP45 staining (Supplementary Fig. 1f). Therefore, APR2 disruption did not affect IMC formation. In TEM micrographs of WT ookinetes, APR is closely adjoined with apical IMC (100%, $n$ = 10) (Fig. 6b, upper panel). However, APR-SPMT moved together posteriorly for a distance, leaving a significant gap between APR and apical IMC in the *Δapr2* ookinetes (100%, $n$ = 12) (Fig. 6b, low panel). APR-SPMT detachment from apical IMC suggests impaired anchorage of IMC-APR after the loss of APR2. The detached APR in the *Δapr2* appeared smaller compared with that in WT ookinetes (Fig. 6b), consistent with the observation for APR2 protein signal in Fig. 5b, e. Moreover, we performed U-ExM combined with NHS-ester dye staining to check the ookinete apical pellicle. Section projection after confocal scanning of NHS signal also revealed the detached and impaired APR-SPMT in the *Δapr2* ookinetes (Fig. 6c).

To see the defected apical anchorage of SPMT in more detail, the parasites from 12-h in vitro culture were SDC-extracted and examined under TEM after negative staining with phosphotungstic acid (NS-TEM), an approach for observing MT cytoskeleton in *T. gondii* and *Plasmodium*[4,35]. In ookinete ghost depleted of pellicle membranes, the SPMTs radiated densely from the apical and arranged in a dome-like pattern in WT parasites (Fig. 6d). Detailed images further revealed that SPMTs were covered outwardly with an amorphous membrane-like layer (Fig. 6d), likely the residue of pellicle membranes after detergent-extraction. Different from WT, a notable gap between SPMT and the outward membrane-like layer was observed in all the *Δapr2* ookinetes (Fig. 6d). Quantification of this phenotype showed that after genetic disruption of APR2, the distance between SPMT and the outward membrane-like layer was remarkably increased (70 ± 22 nm in WT, $n$ = 72; 181 ± 57 nm in *Δapr2*, $n$ = 47) (Fig. 6d). In addition, the overall signals of apical structures in the NS-TEM images were lower in the

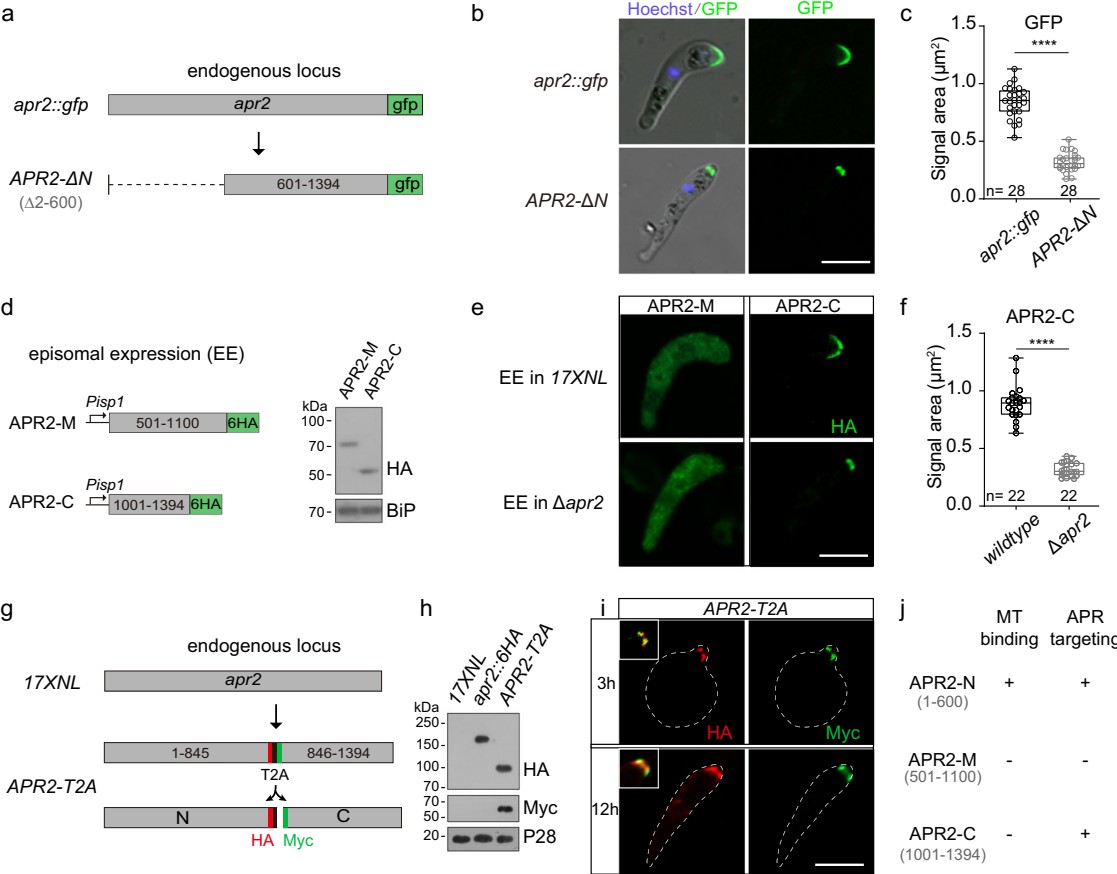

**Fig. 5 | APR2 C-terminal region could localize at APR. a** Diagram of truncation of N-terminal fragment (2–600 aa) in endogenous APR2 in the parental parasite *apr2::gfp*, generating the *APR2-ΔN* strain. **b** Fluorescence of GFP in the *apr2::gfp* and *APR2-ΔN* ookinetes. Scale bars: 5 μm. Two independently performed experiments with similar results. **c** Quantification of GFP signal area in **b**. Boxes show medians with interquartile ranges, whiskers: min to max show all points. *n* = 28 cells examined in both groups. ****P = 3e−24, two-sided Mann–Whitney test. **d** Diagram and confirmation of episomal expression (EE) for HA-tagged APR2-M (501–1100 aa) and APR2-C (1001–1394 aa). Immunoblot confirmed the APR2-M and APR2-C expressed in the *wildtype* ookinetes (1.0 × 10⁶ ookinetes were lysed in each sample). BiP is a loading control. Two independently performed experiments with similar results. **e** IFA of HA-tagged APR2-M and APR2-C episomally expressed in 17XNL and *Δapr2* ookinetes. Scale bars: 5 μm. **f** Quantification of APR2-C IFA signal area in **e**. Boxes

show medians with interquartile ranges, whiskers: min to max show all points. *n* = 22 cells examined in both groups. ****P = 1e−20, two-sided Mann–Whitney test. **g** Diagram of a ribosome-skipping T2A peptide insertion into the endogenous APR2 protein in 17XNL parasite, generating the *APR2-T2A* strain with separated expression of the N- (HA-tagged) and C- (Myc-tagged) parts of APR2. Two independently performed experiments with similar results. **h** Immunoblot of N- and C- parts of APR2 in the *APR2-T2A* ookinetes (1.0 × 10⁶ ookinetes were lysed in each sample). P28 is a loading control. **i** IFA of the HA-tagged N-part and the Myc-tagged C-part of APR2 in the *APR2-T2A* ookinetes from 3- and 12-h in vitro culture. Merged signals were shown in white box, Scale bars: 5 μm. Two independently performed experiments with similar results. **j** Summary of APR2-N, APR2-M, and APR2-C in MT-binding and APR targeting.

*Δapr2* than in WT ookinetes (Fig. 6d), indicating a compromised structure of APR, SPMT, or both after the loss of APR2.

To better track APR in a cell-structure-specific manner, we sought to label APR via the biotin ligase TurboID-based proximity labeling (PL) in the living ookinetes (Supplementary Fig. 6a). The endogenous ARA1, an APR-associated protein of unknown function[26], was used as an APR-biotinylizer by fusing with a TurboID::HA motif in the 17XNL and *Δapr2* parasites, respectively, generating two modified lines *ara1::TurboID* (*Tb-ARA1*) and *ara1::TurboID;Δapr2* (*Tb-ARA1/Δapr2*) (Supplementary Fig. 6a). The fusion ligase protein Tb-ARA1 was localized at APR, overlapping with the biotinylated proteins in biotin-supplemented ookinetes of both *Tb-ARA1* and *Tb-ARA1/Δapr2* parasites (Supplementary Fig. 6b), confirming the biotinylated labeling of APR. Furthermore, PL coupled with U-ExM (PL-U-ExM) for *Tb-ARA1* ookinetes (Fig. 6e) detected a refined 3D structure of APR, a top ring plus approximately 55–60 radiating spines which fit the apical SPMTs in number (Fig. 6f, Supplementary Fig. 6c, d). This PL-U-ExM also enabled visualizing the relative position of APR within the ookinete. In whole-cell 3D graphs, APR closely adjoined with apical IMC in *Tb-ARA1* ookinetes (Fig. 6f). In contrast, a significant gap between APR and apical IMC was observed in

the *Tb-ARA1/Δapr2* ookinetes (Fig. 6f), in agreement with the observation by the super-resolution images in Fig. 6b–d. Compared with the APR in ookinetes of parental parasites, the spines in the detached APR were destroyed in APR2-null ookinetes (Fig. 6f). Quantification analysis of structures revealed a significant reduction in size and surface area of APR (Fig. 6g). Together, these results demonstrated that APR2 is required for both apical anchorage and structural integrity of APR-SPMT during ookinete development.

**Cortical arrangement of SPMT is impaired in the APR2-null ookinetes**

Next, we asked whether apical detachment of APR-SPMT affects the global cortical arrangement of SPMT in the APR2-null ookinetes. We stained the living ookinetes with SiR-Tubulin, a fluorescent probe of MTs. Section projection of the fluorescent signals revealed an even distribution of MTs along the pellicle from anterior to posterior in the WT ookinetes. However, a significantly decreased signal was observed in the anterior two-thirds of the *Δapr2* ookinete (Fig. 7a, b), indicating a perturbed global arrangement of cortical SPMT in the absence of APR2. Furthermore, we stained ookinetes with antibodies against

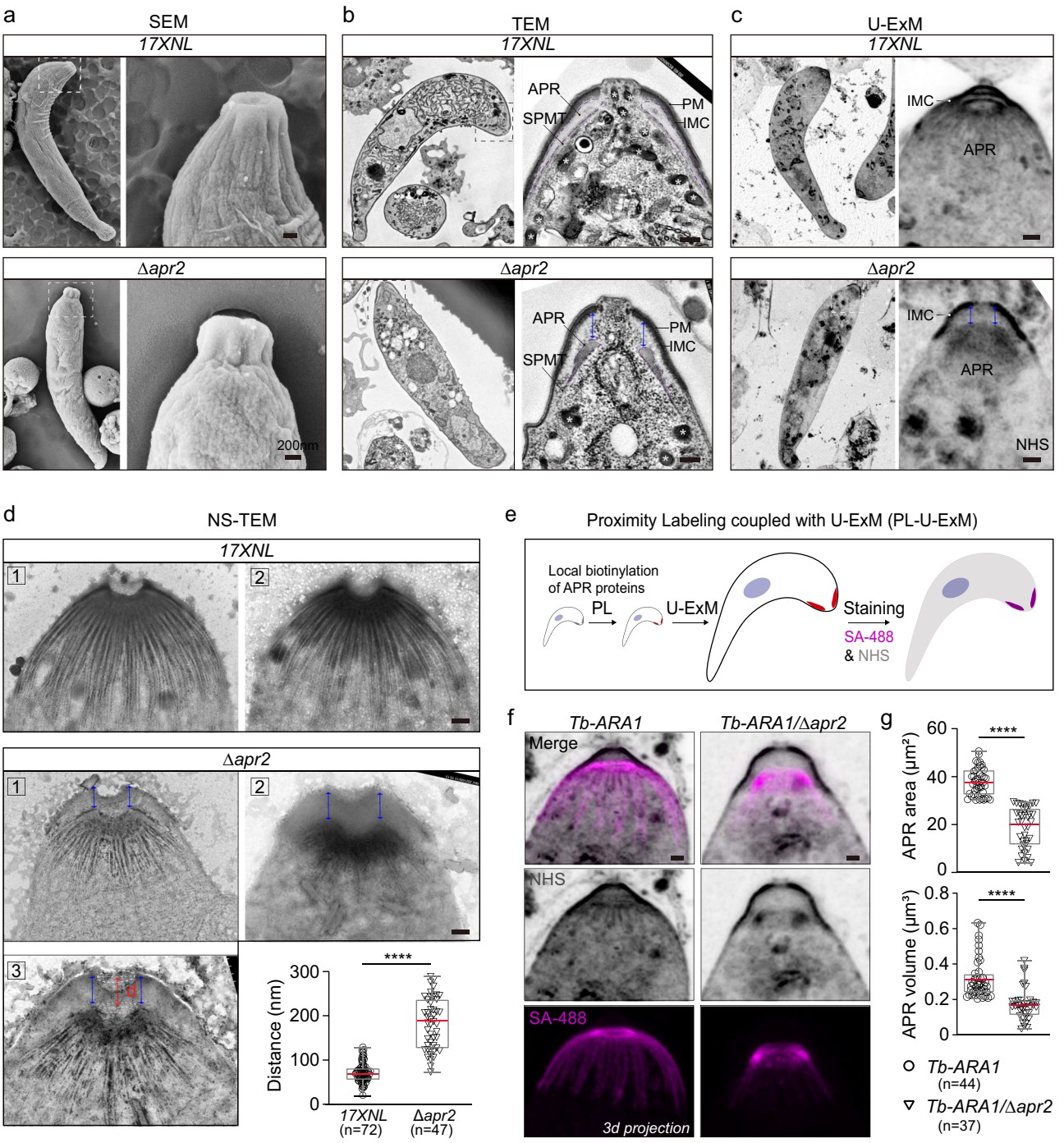

either Tubulin or Tubulin polyglutamylation (PolyE), a posttranslational modification of MT[36]. U-ExM analysis of either Tubulin or PolyE signal detected similar defects of global SPMTs, many of which lost cortical attachment and disorderly scattered in the cell (Fig. 7c, d). To concurrently track the global SPMTs and the APR at a higher resolution, we applied the PL-U-ExM to analyze the biotinylated ookinetes of the *Tb-ARA1* and *Tb-ARA1/Δapr2* parasites after co-staining with anti-Tubulin antibodies and the fluorescent-conjugated streptavidin. Intact APR and cortical SPMTs radiating from APR constituted a crescent-shaped cytoskeleton in the *Tb-ARA1* ookinetes. In contrast, the spines of APR were severely destroyed in the *Tb-ARA1/Δapr2* ookinetes (Fig. 7d). As a result, many of SPMTs lost apical connection with APR and cortical attachment to the pellicle (Fig. 7d).

TEM analysis of the ookinete transversal sections revealed that approximately half of SPMTs were associated with IMC and distributed evenly around the cell periphery of the WT ookinetes (Fig. 7e). In contrast, the number of IMC-associated SPMTs was greatly decreased in *Δapr2* compared to WT ookinetes (Fig. 7e, f). As a consequence, the distance between the adjacent IMC-associated SPMTs changed dramatically, with smaller and larger gaps appearing more frequently in *Δapr2* ookinetes (Fig. 7f). The perturbed IMC association of SPMTs was also supported by the decreased membrane association of α- and β-Tubulin in *Δapr2* ookinetes compared with WT using the detergent extraction-based protein solubility assay (Fig. 7g). These data further demonstrated the role of APR2 in the integrity and apical anchorage of APR-SPMT, which is critical for maintaining global cortical arrangement of SPMTs.

We also noticed a marked decrease of apical micronemes in ookinetes of the *Δapr2* compared to WT, where most of the micronemes are apically localized from the TEM images (Fig. 6b and

**Fig. 6 | APR2 is required for integrity and apical anchorage of APR-SPMT.**
**a** Representative images from scanning electron microscopy (SEM) of 17XNL and
*Δapr2* ookinetes. Ookinete apical (white dashed line in the left panel) was zoomed
in and shown in the right panel. Scale bars: 200 nm. Three independently per-
formed experiments with similar results. **b** Representative images from transmis-
sion electron microscopy (TEM) of 17XNL and *Δapr2* ookinetes. Ookinete apical
(black dashed line in the left panel) was zoomed in and shown in the right panel.
APR, SPMT, PM and IMC are indicated. A gap (blue arrow) between apical IMC and
APR was observed in the mutant parasites. Micronemes were labeled with asterisk.
Scale bars: 200 nm. Three independently performed experiments with similar
results. **c** Representative images from the U-ExM of 17XNL and *Δapr2* ookinetes
stained with the NHS-ester dyes. Ookinete apical (white dashed line in the left
panel) was zoomed in and shown in the right panel. APR and IMC are indicated. A
gap (blue arrow) between apical IMC and APR was observed in the mutant parasites.
Scale bars: 200 nm. Three independently performed experiments with similar
results. **d** Representative images from negative staining TEM (NS-TEM) of 17XNL
and *Δapr2* ookinetes. A gap (blue arrow) between SPMT and apical membrane-like
layer was emerged in the *Δapr2* ookinetes, but not in 17XNL ookinetes. The distance
(d) between SPMT and the apical membrane-like layer was quantified and shown in

the low right panel. Scale bars: 200 nm. n is the number of ookinetes in each group.
Boxes show means (red line) with interquartile ranges, whiskers: min to max show
all points. ****$P = 6e{-}29$, two-sided Mann–Whitney test. Three independently per-
formed experiments with similar results. **e** Flow-chart of Proximity Labeling cou-
pled with U-ExM (PL-U-ExM) for detecting a high-resolution 3D structure of APR
within ookinete. Two modified strains *ara1::TurboID* (*Tb-ARA1*) and *ara1::Turbol-
D;Δapr2* (*Tb-ARA1/Δapr2*) with endogenous ARA1 tagged with a TurboID::HA motif
were generated. APR was labeled via the biotin ligase TurboID-based PL in living
ookinetes after incubation with 50 μM biotin. The ookinetes were expanded and co-
stained with the fluorescent-conjugated streptavidin (SA-488) and the NHS-ester
dye for APR and pellicle imaging. **f** Representative maximum intensity projection
(MIP) images from PL-U-ExM of the *Tb-ARA1* and *Tb-ARA1/Δapr2* ookinetes. APR
(magenta) and pellicle (black) in the ookinete apical were shown. Ookinetes were
expanded by 4×fold. Scale bars: 200 nm. Three independently performed experi-
ments with similar results. **g** Quantification of APR surface area and volume in **f**, *n* is
the number of cells analyzed. Boxes show means (red line) with interquartile ran-
ges, whiskers: min to max show all points. From top to buttom: ****$P = 9e{-}20$ and
****$P = 4e{-}08$, by two-sided Mann–Whitney test.

---

Supplementary Fig. 7a). Disturbed pellicle attachment of SPMTs,
especially the apical SPMTs, may cause microneme mislocalization
from apical to the cytoplasm, which is consistent with the previous
notion of SPMTs in controlling apical localization of micronemes in
*Plasmodium* merozoites and sporozoites[37,38]. To determine whether
the change in ookinete gliding ability could be attributed to a defect in
microneme secretion, we performed an immunoblot assay to quantify
the secreted proteins. The microneme-secreted proteins CTRP, chit-
inase, and WARP were reduced in the ookinete culture supernatants of
*Δapr2* compared to WT, while their overall levels in total ookinete
lysates were comparable (Supplementary Fig. 7b, c). Therefore, APR2
depletion also affects apical localization and secretion of the micro-
nemes due to the pellicle detachment of SPMTs.

### Detection of APR proteome using TurboID proximity labeling

Since the molecular compositions of APR remain elusive in *Plasmo-
dium*, the identification of APR2 makes it feasible to characterize an
APR proteome. We applied the TurboID-based PL to track APR pro-
teome or APR2-interacting proteins in the ookinetes. The endogenous
APR2 was fused with a TurboID::HA motif in the 17XNL parasite, gen-
erating the modified line *apr2::TurboID* (*Tb-APR2* in short) for PL
(Fig. 8a). We also generated a control line *apr2::T2A::TurboID::HA* (*Tb-
CytoI* in short), in which a T2A peptide was inserted to direct cytosolic
expression of TurboID::HA alone under the promoter of *apr2* gene
(Fig. 8a). As expected, the fusion protein Tb-APR2 was localized at APR
while the Tb-CytoI was cytosolic in the ookinetes (Fig. 8b). After
incubation with 50 μM biotin for 3 h at 22 °C, the ookinetes expressing
ligases were co-stained with fluorescence-conjugated streptavidin and
anti-HA antibody. As expected, the biotinylated proteins apically co-
localized with the fusion ligase in *Tb-APR2* ookinetes (Fig. 8b), while the
biotinylated proteins and the ligase were in cytosolic in *Tb-CytoI*
ookinetes (Fig. 8b). Three biological replicates were prepared from the
*Tb-APR2* and *Tb-CytoI* ookinetes, and the streptavidin-affinity purified
proteins from cell extracts were subjected to further proteomic ana-
lysis. Quantitative mass spectrometry yielded 110 enriched proteins
with high confidence in the *Tb-APR2* compared to the *Tb-CytoI* ooki-
netes (Fig. 8c, Supplementary Data 1). APR2, after being cis-biotiny-
lated, was expected in this protein dataset (Fig. 8c). The other hits were
predominantly proteins of unknown function. Notably, the known
APR-associated protein ARA1 was the top hit in the dataset, suggesting
good quality of our PL method.

To further validate the *Tb-APR2/Tb-CytoI* PL results and the
method per se, we generated another two modified lines *Tb-ARA1* and
*Tb-CytoII* for a reverse PL using the ARA1 as the bait (Fig. 8d, e). Pro-
teomic analyses of the *Tb-ARA1/Tb-CytoII* PL obtained another list of
137 high-confidence hits including APR2 (Fig. 8f, Supplementary

Data 2). These hits overlapped extensively with those from the *Tb-
APR2/Tb-cyto1* dataset (Fig. 8g), suggesting that the hits represented
high-probability APR proteins. To test it, the top 10 hits from the
overlapped proteins of two datasets, including ARA1 and 9 other
uncharacterized proteins (named with APRp1 to APRp9), were selected
for detecting their subcellular localization (Fig. 8g). Each candidate
was tagged with a 6HA at the C-terminus and driven by the promoter of
gene *isp1* for episomal expression in the *apr2::gfp* ookinetes. The IFA
showed that all 10 hits (ARA1, and APRp1 to APRp9) displayed clear
apical co-localization with APR2 in the ookinetes (Fig. 8h). Together,
using the APR protein APR2 as the bait, we characterized the first APR
proteome of the ookinete in the *Plasmodium*.

### APR2-APRp2-APRp4 module regulates apical anchorage of APR-SPMT

To better define the mechanism of APR2 in regulating apical anchorage
and integrity of APR-SPMT, we performed a yeast two-hybrid assay
(Y2H) to identify proteins interacting directly with APR2 from the 10
validated APR proteins in Fig. 8h. APR2 was designed as the bait and
the 10 APR-localizing proteins as the pray. Because the full-length
APR2 showed auto-activation in the Y2H experiment, two fragments
APR2$^{F1}$ (aa 1–600) and APR2$^{F2}$ (aa 601–1394) were used as independent
baits (Fig. 9a). The results showed that APR2$^{F2}$ interacted with 3 pray
proteins ARA1, APRp2, and APRp4, while APR2$^{F1}$ did not (Fig. 9a). Each
of the endogenous ARA1, APRp2 and APRp4 proteins was tagged with a
6HA using the CRISPR-Cas9 in both WT and *apr2::gfp* background,
generating 3 single-tagged strains (*ara1::6HA*, *aprp2::6HA*, and *apr-
p4::6HA*) and 3 double-tagged strains (*apr2::gfp;ara1::6HA*, *apr2::gf-
p;aprp2::6HA*, and *apr2::gfp;aprp4::6HA*). The close association
between APR2 and each of ARA1, APRp2, and APRp4 was further
confirmed using the PLA assay, in which the signals were detected at
ookinete apical of the double-tagged strains (Fig. 9b).

To explore the functional relevance of ARA1, APRp2, and
APRp4 in controlling the integrity and apical anchorage of APR-
SPMT, we disrupted each of these genes in WT parasite using the
CRISPR-Cas9, obtaining 3 mutant strains *Δara1*, *Δaprp2*, and
*Δaprp4*. The NS-TEM analysis revealed that the gap between APR-
SPMT and the outward membrane-like layer emerged in *Δaprp2*
and *Δaprp4* mutant ookinetes (Fig. 9c and Supplementary Fig. 8a).
The NS-TEM signals of APR-SPMTs were decreased in *Δaprp2* and
*Δaprp4* compared to WT ookinetes (Fig. 9c and Supplementary
Fig. 8a). In contrast, ARA1 depletion did not affect apical ancho-
rage and integrity of APR-SPMT in the ookinetes (Fig. 9c). The
*Δaprp2* and *Δaprp4* mutants showed normal asexual blood stage
proliferation and gametocyte production. However, both mutants
displayed a moderate decrease in mature ookinete formation

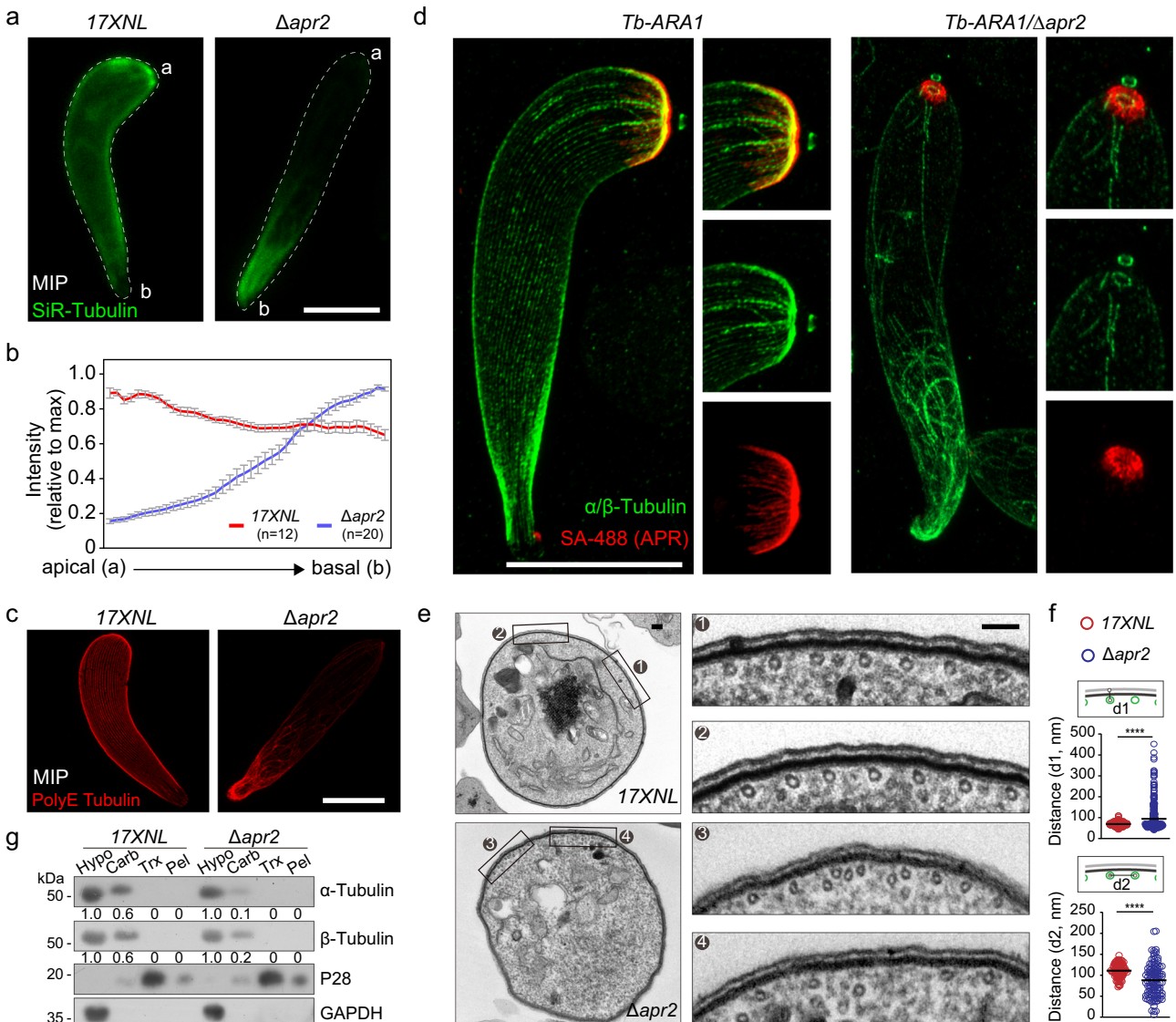

**Fig. 7 | Cortical arrangement of SPMT is impaired in the APR2-null ookinetes.**
**a** Full section projections of fluorescent signal in living 17XNL and Δ*apr2* ookinetes stained with SiR-Tubulin (MT fluorescent probe). Representative maximum intensity projection (MIP) images were shown. a, apical end; b, basal end. Scale bars: 5 µm. Three independently performed experiments with similar results.
**b** Quantification of fluorescent signal in **a** along pellicle from apical to basal. The maximum intensity was set as 1.0 and all signals were normalized. *n* is the number of ookinetes analyzed from two independent experiments. Data are shown means ± SEM. **c** IFA of SPMTs by staining the glutamylated tubulin (PolyE) in the expanded 17XNL and Δ*apr2* ookinetes. Representative maximum intensity projection (MIP) image was shown. Scale bars: 5 µm. Two independently experiments performed. **d** PL-U-ExM of the *Tb-ARA1* and *Tb-ARA1/Δapr2* ookinetes with APR stained with SA-488 (red) and SPMT stained with α- and β-Tubulin antibodies (green) respectively. Representative maximum intensity projection (MIP) images showing SPMT (green) and APR (red) in left panel, and enlarged views of the apical in right panels. Scale bars: 5 µm. Three independently performed experiments with similar results. **e** TEM of ookinete cross sections showing the arrangement of SPMTs underneath IMC in 17XNL and Δ*apr2* parasites. Approximately 60 hollow

SPMTs are associated with IMC, distributing evenly around pellicle in 17XNL ookinetes. Approximately half of SPMTs lost association with IMC in the Δ*apr2* ookinetes. One representative cell in each parasite is shown with two selected areas zoomed in. Scale bars: 100 nm. Three independently performed experiments with similar results. **f** Quantification of distance (d1) between SPMT and PM and distance (d2) between the adjacent IMC-associated SPMTs in ookinetes in **e**. Values are means ± SD for *n* = 647 for d1 and 108 for d2 (17XNL) and *n* = 365 for d1 and 108 for d2 (Δ*apr2*) measurements in 20 cells each group from two independent experiments; From top to buttom: ****$P$ = 2e−21 and ****$P$ = 3e−07, by Kolmogorov–Smirnov test. **g** Protein solubility assay detecting membrane association of α- and β-Tubulin in 17XNL and Δ*apr2* ookinetes 2.0 × 10⁶ ookinetes were lysed in each sample. Cytosolic soluble proteins are in hypotonic buffer (Hypo), peripheral membrane proteins in carbonate buffer (Carb), integral membrane proteins in Triton X-100 buffer (Trx), and insoluble proteins in pellet (Pel). P28 is an integral plasma membrane protein, and GAPDH is a cytosolic soluble protein. The numbers are the relative intensities of band in the immunoblot. Two independent experiments were repeated.

(Fig. 9d). The mature-looking mutant ookinetes lost the crescent shape (Fig. 9e) and had an impaired gliding ability compared to the WT (Fig. 9f). As a consequence, both Δ*aprp2* and Δ*aprp4* mutants produced significantly decreased numbers of midgut oocyst in the infected mosquitoes (Fig. 9g). These defects caused by depletion of either APRp2 or APRp4 were in concordance with the phenotype of the Δ*apr2* mutant.

Given the apparent role of APRp2 and APRp4 in apical anchorage of APR-SPMT, we further investigated their localization dynamics in ookinete development. APRp2 and APRp4 showed apical localization throughout ookinete development (Supplementary Fig. 8b). High-resolution imaging via U-ExM also revealed that APRp2 and APRp4 displayed the ring pattern in early ookinetes and the ring plus spines pattern in mature ookinetes, similar as APR2 (Supplementary Fig. 8c).

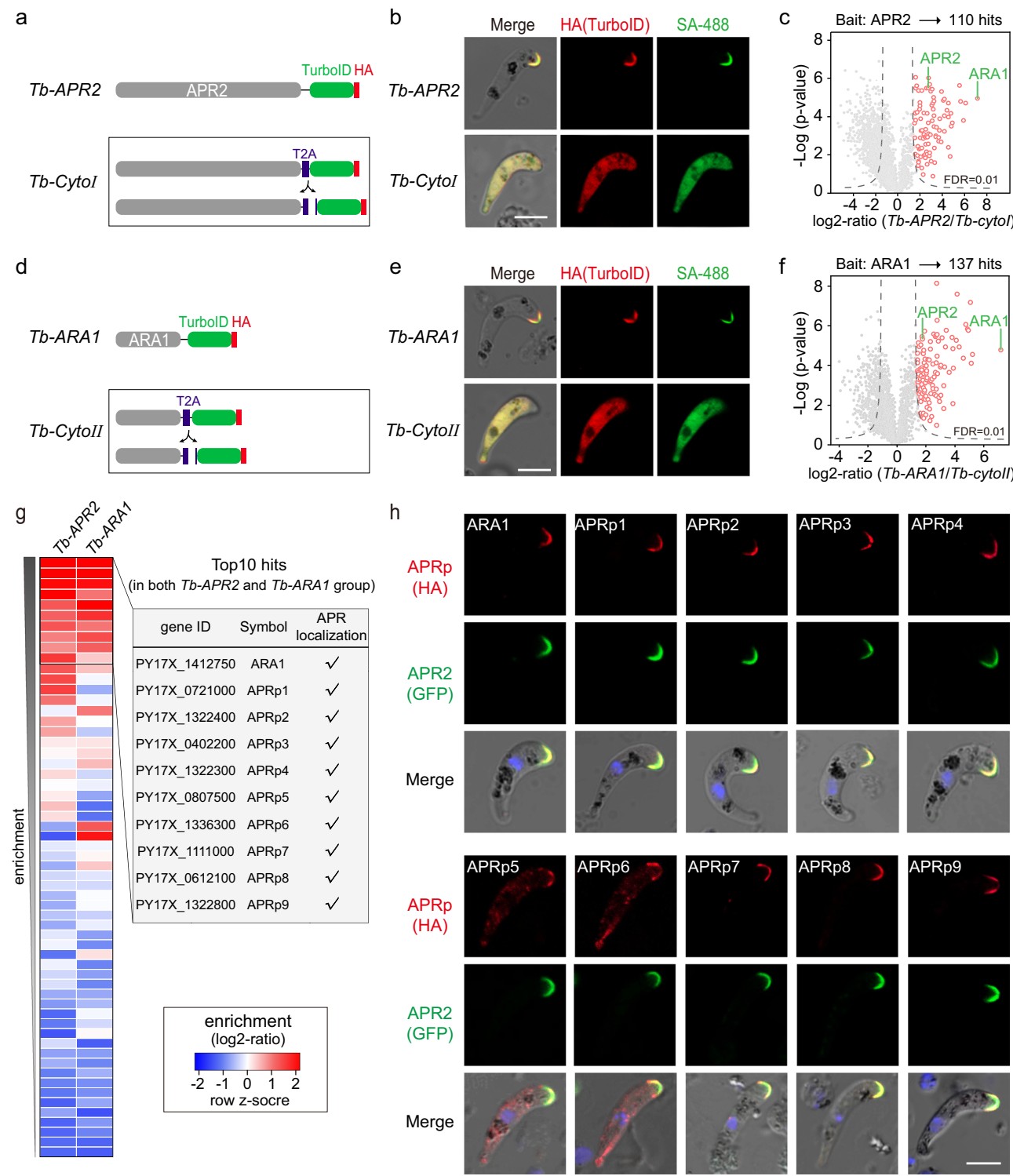

In addition, we detected the protein interaction between APRp2 and APRp4 by the Y2H method (Supplementary Fig. 8d). Together, these results demonstrated that APR2, APRp2, and APRp4 had pairwise interactions with each other and constituted a functional module regulating the integrity of APR and the apical anchorage of APR-SPMT.

We noticed that all the APR2-null ookinetes showed a marked decrease for protein occupancy at APR with smaller signal area of tested proteins (Figs. 5b, e, 6b–d, f, 7d, and Supplementary Fig. 6b), indicating compromised deposition of APR-residing proteins. Based on those observations, we checked the APR localization of APRp2 and APRp4 in APR2-null ookinetes. We deleted the

endogenous *apr2* gene in each of *aprp2::6HA* and *aprp4::6HA* strains, obtaining *aprp2::6HA;Δapr2* and *aprp4::6HA;Δapr2* mutants. IFA showed that APRp2 and APRp4 were still apically localized after APR2 depletion (Supplementary Fig. 8e). However, the IFA signal area of APRp2 and APRp4 was significantly decreased in the APR2 mutants compared to their parental lines (Supplementary Fig. 8e, f). These results for APRp2 and APRp4 in APR2-null ookinete were further confirmed by immuno-EM, with fewer colloidal-gold particles detected in the impaired and detached APR (Supplementary Fig. 8g). Therefore, APR targeting of APRp2 and APRp4 seems not to rely on APR2, but APR2 depletion impairs APR integrity and

**Fig. 8 | Detection of APR proteome using TurboID proximity labeling.**
**a** Schematic of the modified strain used for TurboID ligase-mediated PL of APR proteins in ookinete. Endogenous APR2 was C-terminally tagged with a TurboID::HA motif by CRISPR-Cas9 in 17XNL, generating *Tb-APR2* strain for APR biotinylation. *Tb-CytoI* was generated as a reference strain in which the T2A is inserted between APR2 and TurboID for separated expression of APR2 and TurboID. **b** Co-staining of HA-tagged TurboID ligase (red) and biotinylated proteins (SA-488, green) in *Tb-APR2* and *Tb-CytoI* ookinetes. Ookinetes incubated with 50 μM biotin at 22 °C for 3 h were co-stained with SA-488 and anti-HA antibody. Scale bars: 5 μm. Two independently performed experiments with similar results. **c** Volcano plots illustrating 110 APR2-interacting proteins (red cycle) detected in the *Tb-APR2* versus *Tb-CytoI* ookinetes. See Supplementary Data 1 for dataset. Relative enrichment ratio (x axis) of each protein was calculated by quantifying protein intensity in *Tb-APR2* relative to *Tb-CytoI* (n = 3), *p*-value (y axis) was calculated by two-sided *t* test between two groups. Dashed lines indicated the threshold value of FDR = 0.01. APR2 and ARA1 are highlighted. **d** Schematic of independently modified strain used for PL of APR proteins in ookinete. Endogenous ARA1 was tagged with a

TurboID::HA by CRISPR-Cas9 in 17XNL, generating *Tb-ARA1* strain for APR biotinylation. *Tb-CytoII* was generated as a reference strain in which the T2A is inserted between ARA1 and TurboID. **e** Co-staining of HA-tagged TurboID ligase (red) and biotinylated proteins (SA-488, green) in *Tb-ARA1* and *Tb-CytoII* ookinetes. Scale bars: 5 μm. Two independently performed experiments with similar results. **f** Volcano plots illustrating 137 ARA1-interacting proteins (red cycle) detected in the *Tb-ARA1* versus *Tb-CytoII* ookinetes. See Supplementary Data 2 for dataset. Relative enrichment ratio (x axis) of each protein was calculated by quantifying protein intensity in *Tb-ARA1* relative to *Tb-CytoII* (n = 3), *p*-value (y axis) was calculated by two-sided *t* test between two groups. Dashed lines indicated the threshold value of FDR = 0.01. APR2 and ARA1 are highlighted. **g** Heatmap of APR protein candidates from **c** and **f**. There are 33 hits (FDR < 1% and log2-ratio > 2) detected in both *Tb-APR2/Tb-CytoI* and *Tb-ARA1/Tb-CytoII* PL dataset. The top 10 hits were chosen for further analysis. **h** IFA of APR protein candidates (top 10 hits in **g**) expression in the ookinetes. Each candidate protein was C-terminally fused with a 6HA and episomally expressed in ookinetes of the *ap2::gfp* parasite. Scale bars: 5 μm. Two independently performed experiments with similar results.

---

consequently reduces the overall deposition of APR proteins including APRp2 and APRp4.

## Discussion

The polarized shapes of apicomplexan zoites, essential for parasite invasion and gliding, are determined by the remarkable cytoskeletal architectures of the SPMTs[4,13,39]. How the parasites establish and maintain these polarized architectures of SPMTs is not clear. The APR serves as a unique apical MTOC from which the SPMTs anchor and radiate. However, APR in the apicomplexan zoite, especially in *Plasmodium* ookinete, has remained a mysterious organelle regarding its structure, composition, and function. In this study, we revealed a high-resolution ookinete APR structure with a top ring plus the radiating spines, distinct from the single-ring APR in apicomplexan zoites. In addition, we identified a functional module of APR2-APRp2-APRp4 residing at the APR of the *Plasmodium* ookinetes. Disruption of each protein of this module impaired integrity and apical anchorage of ookinete APR, resulting in detachment of global SPMTs from both apical and pellicle.

Previous ultrastructural studies of the *Plasmodium* ookinetes by TEM have revealed APR an electron-lucent cap-shaped structure, which adjoins with the apical IMC and connects with the minus end of the SPMTs[11,19,20] (Supplementary Fig. 3a). By utilizing the cryoelectron tomography, Josie L Ferreira *et.al* have recently revealed that *Plasmodium* ookinete APR had a near-native structure of two concentric layers with separated "tentacles" on their basal side[20]. Native expansion of parasite samples by U-ExM recently adapted has offered a feasible method for analyzing the apical structures of apicomplexan zoites with unprecedented resolution[14,25]. In this study, the U-ExM unambiguously established that the ookinete APR is structurally composed a top ring plus approximate 60 radiating spines with each spine directly contacting with the minus end of a SPMT (Figs. 3f and 6f, Supplementary Fig. 6c, d). The APR model of ring plus spines fitted the model of ring plus tentacles. Importantly, the APR structure composing a top ring plus spines in ookinete updated the commonly-accepted view of the single-ring APR, which was observed or described in the *Plasmodium* merozoite and sporozoite, as well as the *Toxoplasma gondii* tachyzoite[40]. So far, the spines-containing APR is unique to ookinetes as no similar structure reported in other apicomplexan zoites. Among the apicomplexan zoites, the number of SPMTs in ookinete is much more than that in other zoites: around 60 SPMTs in ookinetes[4,8–12], 16 SPMTs in sporozoites[13], 1–4 SPMTs in merozoites[14–16], and 21–24 SPMTs in tachyzoites[41]. In addition, the SPMTs occupy the whole cell pellicle from apical to basal end in ookinetes, but only extend to the 1/2 to 2/3 length of parasite pellicle in merozoites, sporozoites, and tachyzoites[13,20,35]. Compared to the single-ring APR, the APR with the spines could provide a larger platform for IMC-APR-SPMT connection

and anchorage, fitting ookinete requirement for apical anchoring and stabilizing of larger number and length of SPMTs.

To date, it is unclear for *Plasmodium* APR component proteins. APR2 is the first well-characterized *Plasmodium* APR protein, making it feasible to track the ookinete APR proteome using immunoprecipitation or PL. By implementing quantitative TurboID-based PL using APR2 as the bait as well as the cytosol-expressing TurboID ligase as the reference control, we successfully characterized an inventory of APR proteins in the ookinetes. Collectively, around 110 APR protein candidates were obtained. Notably, out of the top 10 selected candidates, all of them were validated to be APR-localizing in the ookinetes. Certain IMC proteins were also identified in this dataset (Supplementary Data 1 and 2), implying a conceivable close association between IMC and APR. In the future, it is important to classify the APR common proteins shared in all three zoites (merozoite, ookinete and sporozoite) and APR proteins unique to specific zoite. In this study, apical or polarized localization of APR2 was also detected in the merozoite and sporozoite, suggesting that APR2 is an APR common protein. The modified strains *Tb-APR2* and *Tb-CytoI* generated in this study provided a valuable resource for the PL of APR in merozoites and sporozoites, which could delineate the APR common and unique protein components among zoites by comparative proteomics. Strikingly, APR2 is critical for SPMT cytoskeleton in ookinete, but not in merozoite, since viable APR2-null parasites were readily obtained at the asexual blood stage, suggesting that the functional requirement of APR2 for APR is zoite-specific or that the physiological importance of APR is different among zoites.

Ookinete APR is presumably believed as a structure involving at least three critical roles: the IMC-APR connection, SPMT nucleation, and APR-SPMT anchorage. It is reasonable to postulate that parasite has evolved distinct APR protein modules to mediate and orchestrate distinct functions. Based on the facts that APR2 resides at APR and binds directly to SPMTs, we initially hypothesized that APR2 is localized at the APR-SPMT interface and regulates SPMT nucleation or APR-SPMT anchorage. However, the SPMTs were assembled and APR-anchored in the APR2-null ookinetes, suggesting that APR2 is not essential for SPMT nucleating and APR-SPMT anchorage. APR-SPMT anchorage may also depend on multiple weak interactions between SPMTs and several APR-residing MT-associated proteins (MAPs) including APR2. In this scenario, other factors contribute to APR-SPMT anchorage in the absence of APR2. Therefore, APR2 unlikely plays roles in SPMT nucleation or APR-SPMT anchorage. Unexpectedly, APR2 depletion caused prominent defects in APR structural integrity, with most of the radiating spines destroyed, resulting in a significant decrease in the APR surface area. In addition, APR2 depletion also decreased protein deposition at APR. As a consequence, the APR lost connection with IMC through an unknown mechanism, causing the

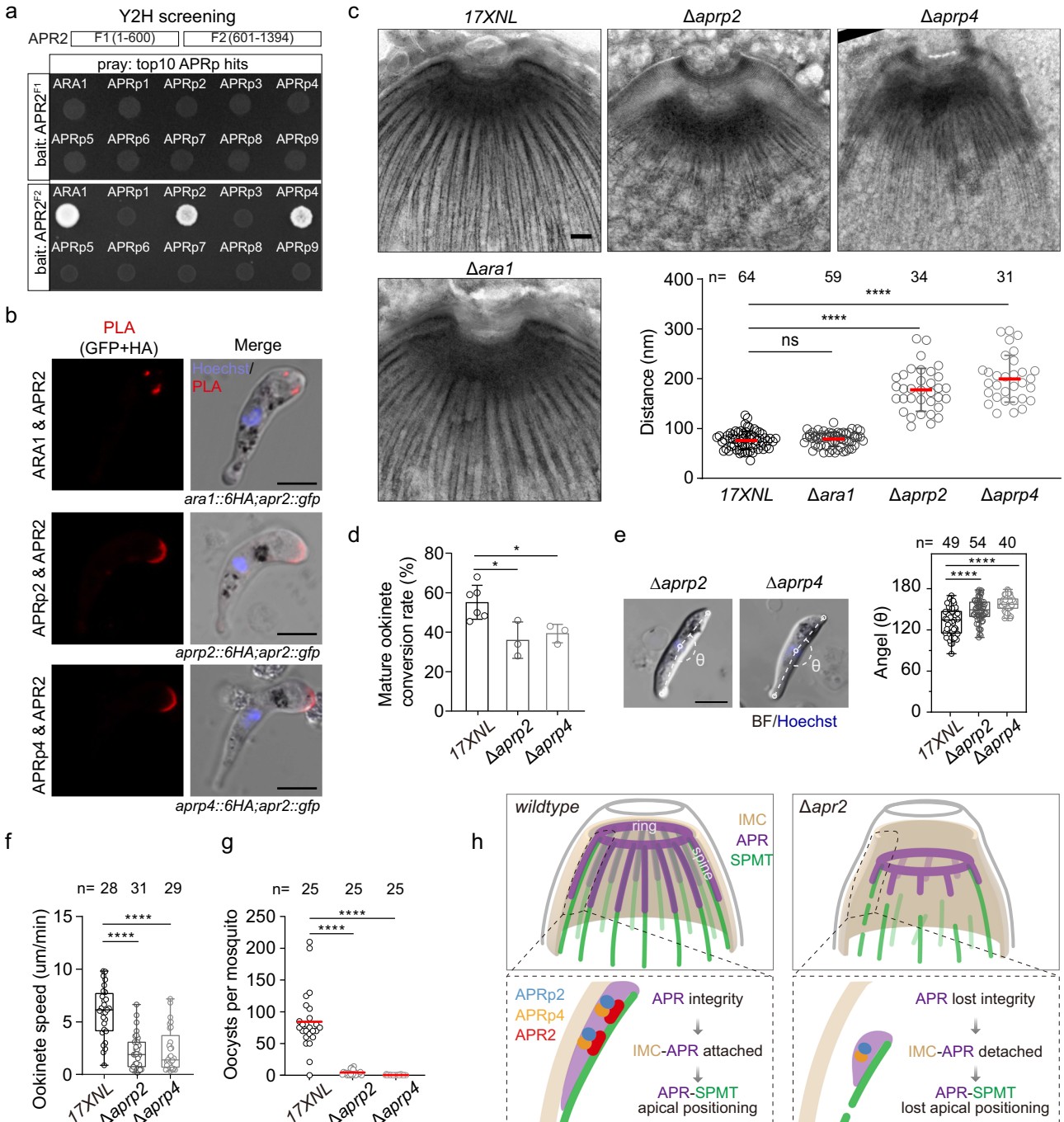

**Fig. 9 | APR2-APRp2-APRp4 module regulates apical anchorage of APR-SPMT.**
**a** Yeast two-hybrid (Y2H) assay for screening the APR2 directly interacting protein. Using two fragments F1 (1–600, aa) and F2 (601–1394, aa) of APR2 as the bait. Yeast expressing paired bait-pray constructs were grown under the restrictive conditions to detect their interactions. Representative results from two independent experiments. **b** PLA detecting protein interaction between APR2 and ARA1, APRp2 and APRp4 in *ara1::6HA;apr2::gfp*, *aprp2::6HA;apr2::gfp* and *aprp4::6HA;apr2::gfp* ookinetes. Scale bars: 5 μm. Two independently performed experiments with similar results. **c** Representative images from negative staining TEM (NS-TEM) of 17XNL, *Δara1*, *Δaprp2*, and *Δaprp4* ookinetes. More images are shown in Supplementary Fig. 8A. Scale bars: 200 nm. Quantification of distance between SPMT and the apical membrane-like layer was shown in lower right panel. *n* is the number of ookinetes in each group from three independent experiments. Values are means ± SD. From left to right: ns *P* = 0.43, ****P* = 2e-29, and ****P* = 6e-33, by two-sided Mann–Whitney test. **d** Mature ookinete conversion rate. Values are means ± SEM (*n* = 3 biological

replicates), From left to right: **P* = 0.02 and **P* = 0.02, by two-sided *t* test. **e** Ookinete cell shape. θ is the angle between the two lines of nucleus-apical and nucleus-basal as illustrated in Fig. 1k. Quantitation of θ in right panel. *n* is the number of ookinetes tested. Boxes show medians with interquartile ranges, whiskers: min to max show all points. From left to right: ****P* = 2e-05 and ****P* = 3e-10, by two-sided Mann–Whitney test. Scale bars: 5 μm. Two independently performed experiments with similar results. **f** Ookinete gliding motility measured using the in vitro Matrigel-based assay. *n* is the number of ookinetes tested. Boxes show medians with interquartile ranges, whiskers: min to max show all points. From left to right: ****P* = 2e-09 and ****P* = 7e-08, by two-sided Mann–Whitney test. **g** Midgut oocysts count in mosquito 7 dpi. *n* is the number of mosquitoes dissected. Red horizontal lines show mean value of oocyst numbers. From left to right: ****P* = 2e-11 and ****P* = 4e-20, by two-sided Mann–Whitney test. **h** Proposed model showing the module of APR2-APRp2-APRp4 in regulating the stabilization of APR and apical anchorage of APR-SPMT.

APR-SPMT as a whole detached from IMC (see the proposed working model in Fig. 9h and Supplementary Fig. 9). Our data indicate that APR2 depletion impaired APR structure integrity, but this study does not further address the mechanism for the IMC-APR disconnection. In future work, it is needed to elucidate the proteins that are localized at the IMC-APR interface and are responsible for the IMC-APR connection. Consistent with the phenotype caused by APR2 deficiency, depletion of either APRp2 or APRp4, two APR-residing proteins directly interacting with APR2, caused similar defects in the structural integrity and apical anchorage of APR-SPMT. These results suggest that APR2, APRp2, and APRp4 act as APR-stabilizing factors to maintain the structural integrity of APR and apical anchorage of APR-SPMT and, hence, support SPMT cytoskeleton during ookinete morphogenesis (Fig. 9h and Supplementary Fig. 9). In addition, we analyzed the conservation of amino acid sequences for the 11 APR proteins (APR2, ARA1, and APRp1-p9) among *Plasmodium* species (including three human malaria parasites and three rodent malaria parasites). Except APRp3, other proteins (APR2, ARA1, APRp1-p2, and APRp4-p9) show variable degree of homology with the orthologs among the *Plasmodium* parasites. These results suggest the conserved mechanism for APR2-regulated APR stability and apical anchorage of SPMT in the *Plasmodium* (Supplementary Fig. 10a, b).

Tubulins (α- and β-isoforms), the core block proteins of MTs, are highly conserved in eukaryotes including *Plasmodium*. Purified *Plasmodium* MTs also showed in vitro dynamics similar to the mammalian MTs[42]. However, the SPMTs in the *Plasmodium* zoites are unique in spatial structure. In addition, the SPMTs are extremely stable as they are resistant to detergent-extraction and cold treatment[8,43]. Therefore, the divergence in the structure and biochemical/biophysical properties between *Plasmodium* SPMTs and conventional MTs should be determined by the unique MAPs, but not by Tubulins themselves. It is reasonable to speculate that certain APR proteins may function as MAPs, but none has yet been identified. In this study, we established that APR2 binds to and stabilizes MTs in both the heterologous mammalian expression and the in vitro assays. In future work, the MT depolymerization agents, like trifluralin or colchicine, could be used to verify the SPMT-stabilization activity of APR2 in WT and Δ*apr2* ookinetes. To the best of our knowledge, APR2 is the first experimentally verified MAP for the *Plasmodium* SPMTs. In APR2-depleted ookinetes, the apical and anterior parts of SPMTs were prevalently shortened, consistent with their role for MT-stabilization. We mapped the fragment within APR2 for MT-binding and found that only APR2-N fragments containing the N-terminal 200 amino acid residues (N200) retained the MT-binding activity (Fig. 4a, d, Supplementary Fig. 4a). However, no canonical MT-binding domain or motif was detected in the APR2. Sequence analysis revealed positively charged amino acid residues (lysine and arginine) highly enriched within APR2-N, but not within APR2-M and APR2-C. Within the N200, there is the highest intensity of lysine and arginine. Certain MAPs with positively charged regions could bind to MT by associating with negatively charged C-terminal ends of Tubulin[44]. Therefore, APR2 is a *Plasmodium* MT-binding protein, and the part of N200 is likely a core region for MT-binding. Interestingly, we also observed that APR2 has a preferential binding with the curved MT. This feature likely contributes to the specific binding of APR2 with apical SPMTs, which displays highest curvature at apical along the ookinete pellicle (Supplementary Fig. 5d).

Besides supporting zoite morphogenesis and zoite shape, SPMTs had been suggested to act as the tracks for movement of apical secretory vesicles or as the scaffold for tethering vesicles. In ookinete, micronemes are the principle apical secretory vesicles which secret proteins essential for parasite gliding and invasion via exocytosis through the apex gateway. Micronemes have been proposed to be transported from ER-Golgi near the nucleus to the apical along the SPMTs. This model is favored by the results showing loss of apical position of micronemes due to the pellicle detachment of apical

SPMTs in the APR2-null ookinetes (Fig. 7a, c, d). Consistent with it, the secretion of microneme proteins were consequently reduced in the Δ*apr2* ookinetes compared to WT. Reduced microneme secretion was not due to any obvious perturbation of microneme formation, as the microneme protein content was not affected in ookinetes after APR2 depletion.

## Methods

### Mice and mosquitoes usage and ethics statement

All animal experiments were performed in accordance with approved protocols (XMULAC20140004) by the Committee for Care and Use of Laboratory Animals of Xiamen University. Female ICR mice (5 to 6 weeks old) were housed in Animal Care Center of Xiamen University (at 22–24 °C, relative humidity of 45–65%, a 12 h light/dark cycle) and used for parasite propagation, drug selection, parasite cloning, and mosquito feedings. The larvae of malaria mosquito *Anopheles stephensi* (*Hor* strain) were reared at 28 °C, 80% relative humidity, and a 12-h light/12-h dark condition in a standard insect facility. Adult mosquitoes were supplemented with 10% (w/v) sugar solution and maintained at 23 °C.

### Plasmid construction and parasite transfection

Parasite CRISPR-Cas9 plasmid pYCm was used for the gene editing[45]. To construct vectors for gene deletion or truncation, we amplified the 5′- and 3′- flanking sequences (500 to 600 bp) at the designed deletion region of target genes as homologous templates using specific primers listed in Supplementary Table 1 and inserted them into the specific restriction sites in pYCm. To construct vectors for gene tagging, we amplified the 5′- and 3′- flanking sequences (500 to 600 bp) at the designed insertion site of target genes as homologous templates. DNA fragments encoding 6HA, 4Myc, GFP and T2A (a ribosomal skipping peptide) were inserted between the homologous templates in frame with the coding sequence of target gene. For each modification, at least two small guide RNAs (sgRNAs) were designed using the online program EuPaGDT. Paired oligonucleotides (Supplementary Table 1) for sgRNA were denatured at 95 °C for 3 min, annealed at room temperature for 5 min, and ligated into pYCm. For parasite electroporation, parasite-infected red blood cells were electroporated with 5 μg plasmid DNA using Lonza Nucleofector. Transfected parasites were immediately intravenously injected into a naïve mouse and exposed to pyrimethamine (6 mg/mL) provided in mouse drinking water 24 h after transfection.

### Genotyping of genetic modified parasites

All modified or transgenic parasites were generated from the *P. yoelii* 17XNL strain or 17XNL-derived parasite lines (Supplementary Table 2). 10 μL parasite-infected blood samples were collected from the infected mice tail vein and red blood cells were lysed using 1% saponin in PBS. Parasite cells were spun down by centrifugation at 13,000 g for 5 min and pellets were washed twice with PBS and boiled at 95 °C for 10 min followed by a centrifugation at 13,000 g for 5 min. Supernatants containg parasite genomic DNAs were subjected for genotyping. For each gene modification, both the 5′ and 3′ homologous recombination events were detected by diagnostic PCR, confirming successful integration of the homologous templates. Parasite clones with targeted modifications were obtained by limiting dilution. At least two clones of each gene-modified parasite were used for phenotypic analysis. Modified parasite clones subject for additional modification were negatively selected to remove pYCm plasmid. Each naïve mouse infected with the pYCm plasmid-carrying parasites was exposure to 5-Fluorouracil (5-FC, Sigma-Aldrich, cat#F6627) in mouse drinking water (2.0 mg/mL) for 6–8 days. After negative selection with 5-FC, two pairs of pYCm-specific primers are used for surveying residual plasmids in the parasites. All PCR primers used in this study are listed in the Supplementary Table 1.

## Parasite intraerythrocytic asexual proliferation in mouse

Parasite proliferation rates in asexual blood stage were determined in mice injected intravenously with $1.0 \times 10^5$ parasites. At least 4 ICR mice were included in each group. Parasite growth was monitored by Giemsa-stained thin blood smears every two days from day 2 to 16 post infection. The parasitemia was calculated as the ratio of parasitized erythrocytes over total erythrocytes.

## Gametocyte induction in mouse

ICR mice were injected intraperitoneally with phenylhydrazine (80 mg/g mouse body weight). Three days after treatment, the mice were infected intravenously with $5.0 \times 10^6$ parasites. Three days post infection, male and female gametocytes were checked via Giemsa-stained thin blood smears. Gametocytemia was calculated as the ratio of male or female gametocyte over parasitized erythrocyte. All experiments were repeated three times independently.

## In vitro ookinete culture and purification

Mouse blood samples carrying 6–10% gametocytemia were collected and immediately added to ookinete culture medium (RPMI 1640, 10% FCS, 100 μM XA (Sigma-Aldrich, cat#D120804), 25 mM Hepes; pH 8.0, 100 μg/mL streptomycin, 100 U/mL penicillin). The gametocytes were cultured at 22 °C for 12–15 h to allow gametogenesis, fertilization, and ookinete differentiation. Ookinetes formation was evaluated based on cell morphology in Giemsa-stained thin blood smears. Ookinete conversion rate was calculated as the number of ookinetes (from stage I to V) over that of female gametocytes. Mature ookinete conversion rate was calculated as the number of crescent-shaped mature ookinete (stage V) over that of total ookinetes (from stage I to V). Ookinetes were purified using Nycodenz density gradient centrifugation as described previously[46]. Cultured ookinetes were collected by a centrifugation at 500 g for 5 min. Parasite pellets were resuspended with 7 mL PBS and transferred onto the top of 2 mL of 63% Nycodenz (Axis-shield, cat#66108-95-0) in a 15 mL Falcon tube. After centrifuging at 1000 g for 20 min, the ookinetes enriched at the interface layer were collected from the Falcon tube. Purity of ookinetes was examined by hemocytometer analysis. Ookinetes with more than 80% purity were used for further experiments.

## Parasite infection and transmission in mosquito

Thirty female *Anopheles stephensi* mosquitoes in one cage were allowed to feed on one anesthetized mouse carrying 6–10% gametocytemia for 30 min. For midgut oocyst counting, mosquito midguts were dissected on day 7 or 8 post blood feeding and stained with 0.1% mercurochrome for oocyst observation. For salivary gland sporozoite counting, mosquito salivary glands were dissected on day 14 post blood feeding, and the average number of sporozoites per mosquito was calculated. For mice infection with sporozoite, 15–20 infected mosquitoes on day 14 post blood feeding were allowed to bite one anesthetized naïve mouse for 30 min. Transmission capability of parasites from mosquito to mouse was monitored daily by Giemsa-stained blood smears for 12 days.

## Ookinete microneme secretion assay

Microneme secreted proteins were examined in in vitro ookinete culture supernatants as previously reported[47]. $5.0 \times 10^6$ purified ookinetes were incubated in 200 μL PBS at 22 °C to allow microneme protein secretion. After 6 h incubation, the supernatants were collected by a centrifugation at 750 g for 3 min, filtered through 0.45 μm filter (Millipore, cat#SLHP033RS). Equal volume of $2 \times$ Laemmli sample buffer was added. All samples were boiled at 95 °C for 10 min and centrifuged at 12,000 g for 5 min. Equal volume of supernatants from each parasite group were used for immunoblot analysis.

## Ookinete gliding assay

All procedures were performed in a temperature-controlled room at 22 °C. 20 μL of the suspended ookinete cultures were mixed with 20 μL of Matrigel (BD Biosciences, cat#356234) on ice. The ookinete and Matrigel mixtures were transferred onto a slide, covered with a coverslip, and sealed with nail varnish. The slide was rest for 30 min before observation under microscope. After tracking a gliding ookinete under microscope, time-lapse videos (1 frame per 20 s, for 20 min) were taken to track ookinete movement using a Nikon ECLIPSE E100 microscope fitted with an ISH500 digital camera controlled by ISCapture v3.6.9.3N software (Tucsen). Ookinete motility speeds were calculated with Fiji software using the MtrackJ plugin[48].

## Protein transient expression in ookinetes

The DNA encoding the target proteins with appropriate 5′-UTR and 3′-UTR regulatory regions was inserted into the pL0019-derived vector with human *dhfr* marker for pyrimethamine selection. 10 μg plasmid DNA was used in one electroporation and transfected parasites were selected with pyrimethamine (70 μg/mL) for 7 days. Phenylhydrazine-pretreated mouse were infected with $5.0 \times 10^6$ parasites and applied with pyrimethamine-selection for additional 3 days for plasmid maintenance and gametocyte production. Gametocytes were collected from infected mice containing 6–10% gametocytemia and used for in vitro ookinete culture and protein expression examination.

## DNA content measurement of ookinete

To evaluate nuclear DNA content changes during zygote to ookinete development post fertilization, parasites from 0 and 4 h in vitro ookinete culture were fixed using 4% paraformaldehyde for 20 min, rinsed twice with PBS, and blocked with 5% BSA solution in PBS for 1 h. Parasites were then incubated with anti-P28 antibody for 1 h and washed twice with PBS. Parasites were then incubated with fluorescent conjugated secondary antibodies for 1 h and followed by three washes with PBS. Parasites were then stained with DNA dye Hoechst 33342 (Thermo Fisher Scientific, cat#23491-52-3) for 10 min and mounted in a 90% glycerol solution. Female gametocytes (P28-negative) and female gametes and zygotes (P28-positive) were measured for the Hoechst 33342 signal. Images were captured using identical settings under a ZEISS LSM 880 confocal microscope. Fluorescent singal intensity was ZEN Microscropy Software from ZEISS (https://www.zeiss.com/microscopy/int/products/microscope-software/zen.html).

## Antibodies and antiserum

The primary antibodies used were: rabbit anti-HA (Cell Signaling Technology (CST), cat#3724S, 1:1,000 for immunoblotting (IB), 1:500 for immunofluorescence (IF), 1:500 for immunoprecipitation (IP)), mouse anti-HA(CST, cat#2367S, 1:500 for IF), rabbit anti-Myc (CST, cat#2272S, 1:1,000 for IB), mouse anti-Myc (CST, cat#2276S, 1:500 for IF), mouse anti-α-tubulin II (Sigma-Aldrich, cat#T6199, 1:1,000 for IF, 1:1,000 for IB), mouse anti-β-tubulin (Sigma-Aldrich, cat#T5201, 1:1,000 for IF, 1:1,000 for IB), rabbit anti-Polyglutamate chain (PolyE) (AdipoGen, cat#AG-25B-0030, 1:1,000 for IF), rabbit anti-GFP (Abcam, cat#ab6556, 1:2,000 for IF, 1:2,000 for IB, 1:1,000 for IP) and mouse anti-GAPDH (Servicebio, cat#GB12002, 1:1,000 for IB). The secondary antibodies used were as follows: HRP-conjugated goat anti-rabbit IgG (Abcam, cat#ab6721, 1:5,000 for IB), HRP-conjugated goat anti-mouse IgG (Abcam, cat#ab6789, 1:5,000 for IB), Alexa 555 goat anti-rabbit IgG (Thermo Fisher Scientific, cat#A21428, 1:1,000 for IF), Alexa 488 goat anti-rabbit IgG (Thermo Fisher Scientific, cat#A31566, 1:1,000 for IF), Alexa 555 goat anti-mouse IgG (Thermo Fisher Scientific, cat#A21422, 1:1,000 for IF), and Alexa 488 goat anti-mouse IgG (Thermo Fisher Scientific, cat#A11001, 1:1,000 for IF). The anti-serums, including the rabbit anti-P28 (1:1,000 for IB, 1:1,000 for IF), rabbit anti-BiP (1:1,000 for IB), rabbit anti-enolase (1:1,000 for IB), rabbit anti-GAP45

(1:1,000 for IF), rabbit anti-WARP (1:1,000 for IB), rabbit anti-CTRP (1:1,000 for IB) and rabbit anti-Chitinase (1:1,000 for IB).

## Immunofluorescence assay

Purified parasites were fixed using 4% paraformaldehyde and transferred to a Poly-L-Lysine coated coverslip. Fixed cells were permeabilized with 0.1% Triton X-100 solution in PBS for 10 min, blocked in 5% BSA solution in PBS for 60 min at room temperature, and incubated with the primary antibodies diluted in 3% BSA-PBS at 4 °C for 12 h. The coverslip was incubated with fluorescent conjugated secondary antibodies for 1 h. Cells were stained with Hoechst 33342, mounted in 90% glycerol solution, and sealed with nail varnish. All images were acquired and processed using identical settings on Zeiss LSM 880 and LSM 980 confocal microscopes.

## Live cell imaging

Parasites expressing GFP-fused proteins were collected in 200 μL PBS, washed twice with PBS and stained with Hoechst 33342 at room temperature for 10 min. After centrifugation at 500 g for 3 min, the parasites pellets were re-suspended in 100 μL of 3% low melting agarose (Sigma-Aldrich, A9414), and transferred evenly on the bottom of a 35-mm culture dish. Parasites were placed at room temperature for 15 min and imaged using a Zeiss LSM 880 confocal microscope with the 63×/1.40 oil objective.

## Fluorescence recovery after photobleaching (FRAP) assay

A laser pulse was used to bleach 80–90% of the fluorescence at an APR rectangle region of ookinete in parasite strains *apr2::gfp* and *gfp* which are expressing the GFP-fused APR2 and GFP alone, respectively. The recovery of fluorescence in the apical rectangle region was monitored. Images were taken in a time-series mode to record the fluorescent signal intensities using a Zeiss LSM 980 confocal microscope. The time-series parameters for acquiring images were set as follows: once scanning every 0.5 s within a period of 30 s for tracking GFP dynamics; while once scanning every 20 s within a period of 300 s for tracking APR2-GFP dynamics. More than 20 ookinetes were analyzed for each group independently.

## Protein extraction and immunoblot

Protein extracts from the asexual blood stage parasites, gametocytes, zygotes, retorts, and ookinetes were lysed in RIPA buffer buffer (0.1% SDS, 1 mM DTT, 50 mM NaCl, 20 mM Tris-HCl; pH 8.0) (Solaribio, cat#R0010) supplemented with protease inhibitor cocktail (Medchem Express, cat#HY-K0010) and PMSF (Roche, cat#10837091001). After ultrasonication, the extracts were incubated on ice for 30 min followed with centrifugation at 12,000 g for 10 min at 4 °C. Clarified supernatant was mixed with same volume of 2x protein sample buffer, boiled at 95 °C for 5 min, and cooled at room temperature. After SDS-PAGE separation, samples were transferred to PVDF membrane (Millipore, cat#IPVH00010). The membrane was blocked with 5% skim milk, probed with primary antibodies for 1 h at room temperature, rinsed 3 times with TBST, and incubated with HRP-conjugated secondary antibodies. Followed by three washes with TBST, the membrane was visualized with enhanced chemiluminescence detection (Advansta,cat#K12045-D10).

## Protein immunoprecipitation

Parasites were lysed in the IP buffer A (50 mM HEPES pH 7.5, 150 mM NaCl, 1 mM EDTA, 1 mM EGTA, 1% Triton X-100, 0.1% sodium deoxycholate) with protease inhibitor cocktail and PMSF. Human 293 T cells were lysed in the IP buffer B (25 mM pH 7.5 Tris-HCl, 150 mM NaCl, 1 mM EDTA, 1 mM EGTA, 1% Triton X-100, 10% glycerol) with protease inhibitor cocktail and PMSF. 1 mL of total cell lysates were incubated with 1 μg primary antibodies (rabbit anti-HA or rabbit anti-GFP) overnight. Protein aggregates were removed by centrifugation at 20,000 g

for 10 min, and protein A/G beads (1:50) were added to the lysates and mixed on a vertical rotating mixer for another 3 h. The beads were washed with IP buffer A or IP buffer B for three times at 4 °C, and then mixed with an equal volume of 2 × SDS sample buffer for protein elution. All samples were boiled at 95 °C for 10 min and centrifuged at 12,000 g for 5 min. Equal volume of supernatants from each sample were used for immunoblotting.

## Extraction of ookinete pellicle cytoskeleton

Extraction of ookinete pellicle cytoskeleton was performed as previously described[30]. Approximately $3.0 \times 10^6$ purified ookinetes were lysed in 200 μL 0.5 mM sodium deoxycholate (SDC) detergent for 5 min at room temperature. Followed by centrifugation at 800 g for 8 min, the pellet fractions containing ookinete cytoskeleton were collected for further experiments.

## Scanning electron microscopy

We used the scanning electron microscope (SEM) to observe the ookinete morphology. Purified ookinetes were fixed with 2.5% glutaraldehyde in 0.1 M phosphate buffer at 4°C overnight, rinsed three times with PBS, and fixed with 1% osmium tetroxide for 2 h. Fixed cells were dehydrated using a graded acetone series, $CO_2$-dried in a critical-point drying device, and gold-coated in a sputter coater as detailed previously[49]. The samples were imaged using a SUPRA55 SAPPHIRE Field Emission Scanning Electron Microscope.

## Transmission electron microscopy

Purified ookinetes were fixed with 2.5% glutaraldehyde in 0.1 M phosphate buffer at 4 °C overnight and processed as previously described[4]. Samples were rinsed three times with 0.1 M phosphate buffer, post-fixed with 1% osmium acid for 2 h, and rinsed three times with 0.1 M phosphate buffer, and dehydrated with concentration gradient ethanol. After embedding and slicing, thin sections were stained with uranyl acetate and lead citrate before imaging. For observation of ookinete cytoskeleton in negative staining TEM (NS-TEM), purified ookinetes were treated with 0.5 mM deoxycholate at room temperature for 5 min. The detergent-resistant cytoskeletons of ookinetes were collected by centrifugation at 800 g for 10 min, resuspended in distilled water, and absorbed on the surface of copper grids for 10 min. Grids were stained with 1% aqueous phosphotungstic acid (pH 7.2) for 30 s and air-dried at room temperature. Parasite images were captured by Hitachi HT-7800 electron microscope.

## Immunoelectron microscopy (Immuno-EM)

Purified ookinetes were fixed in 0.1 M phosphate buffer (pH 7.4) containing 2% paraformaldehyde and 0.1% glutaraldehyde. Samples were dehydrated in ethanol and embedded in LR Gold resin (Electron Microscopy Sciences, cat#14381-UC). Ultrathin sections were blocked with 1% BSA for 10 min, incubated with anti-GFP or anti-HA antibodies, and then with goat anti-rabbit IgG conjugated to gold particles of 15 nm diameter (Leading Biology, cat#AB-0295G-Gold) diluted in a blocking buffer. Finally, the sections were fixed with 2.5% glutaraldehyde for 10 min and stained with 1% uranyl acetate. The samples were examined and imaged by a Hitachi HT-7800 electron microscope.

## Protein solubility assay

About $2.0 \times 10^6$ purified ookinetes were prepared for solubility assay as described with minor optimizations[4]. Different fractions were sequentially extracted by specified buffers. Parasites were lysed in 200 μL hypotonic buffer (10 mM HEPES, 10 mM KCl, pH 7.4), frozen and thawed twice (−80 °C to 37 °C). The lysates were centrifuged at 20,000 g for 5 min at 4 °C and the supernatants containing cytosolic soluble proteins (Hypo) were collected. The pellet after hypotonic lysis was rinsed with 1 mL ice-cold PBS, resuspended in 200 μL freshly

prepared carbonate buffer (0.1 M $Na_2CO_3$ in deionized water), kept on ice for 30 min, then centrifuged at 20,000 g for 5 min at 4 °C. The supernatants containing peripheral membrane proteins (Carb) were collected. The pellet after carbonate extraction was rinsed with 1 mL of ice-cold PBS and suspended in 200 μL freshly prepared Triton X-100 buffer (1% Triton X-100 in deionized water) and kept on ice for another 30 min, then centrifuged at 20,000 g for 5 min at 4 °C. The supernatants containing integral membrane proteins (Trx) were collected. The final pellet (P) including insoluble proteins and non-protein materials was resolubilized in 1 × Laemmli sample buffer. Equal volume of 2 × Laemmli sample buffer was added into the Hypo/Carb/Trx fractions, respectively. All samples were boiled at 95 °C for 10 min and centrifuged at 12,000 g for 5 min. Equal volume of supernatants from each sample were used for immunoblotting. All buffers used for solubility assay contained protease inhibitor cocktail. If not otherwise indicated, all steps were carried out on ice.

### Mammalian cell culture, transfection and protein purification
HEK293T cells were maintained in Dulbecco's modified Eagle's medium (DMEM) supplemented with 10% fetal bovine serum (FBS), 100 U/mL penicillin, and 100 μg/mL streptomycin at 37 °C in a humidified incubator containing 5% $CO_2$. TurboFect transfection reagent (Thermo Fisher Scientific, cat#R0532) was used for cell transfection. Total DNA for each plate was adjusted to the same amount by using relevant empty vector. Transfected cells were harvested at 36 h after transfection for further analysis. MRC5 cell were cultured in DMEM/F12 (Biosharp, cat#BL305A) supplemented with 10% FBS, 100 U/mL penicillin, and 100 μg/mL streptomycin and kept at 37 °C in 5% $CO_2$. FuGENE6 (Promega, E2691) was used to transfect plasmids into MRC5 cells for immunofluorescence assay and live-cell imaging. For protein purification in HEK293T cells, pTT5 vector was used for expression of APR2 fragments. HEK293T cells grown on 15-cm dishes were transfected with 20–30 μg DNA per dish. 36 h post transfection, the transfected cells were collected with cold PBS (4 °C, 10 ml for each 15-cm dish) into 15-mL falcon tubes. Cells were centrifuged at 300 g, 4 °C for 10 min to remove the supernatant, and lysed in 900 μL lysis buffer (50 mM Hepes, 300 mM NaCl, and 0.5% Triton X-100, pH 7.4) containing protease inhibitors (Roche) for 10 min on ice. Cell lysate was centrifuged at 14,000 g, 4 °C for 20 min, and the supernatant was incubated with 60~100 μL StrepTactin beads (GE Healthcare, cat#28-9356-00) at 4 °C for 45 min. After removal of the supernatant by centrifuging at 500 g, 4 °C for 1 min, beads were washed four times with 1 mL lysis buffer and twice with 1 mL wash buffer A (50 mM Hepes, 150 mM NaCl, and 0.01% Triton X, pH 7.4). Proteins were eluted with 60~100 μL elution buffer (50 mM Hepes, 150 mM NaCl, 0.01% Triton X-100, and 2.5 mM desthiobiotin, pH 7.4), snaped frozen, and stored at −80 °C.

### Ultrastructure expansion microscopy (U-ExM)
Purified ookinetes were sedimented on a 15 mm round poly-D-lysine (Sigma-Aldrich, cat#A-003-M) coated coverslips for 10 min. To add anchors to proteins, coverslips were incubated for 5 h in 0.7% formaldehyde (FA, Sigma-Aldrich, cat#F8775)/1% acrylamide (AA, Sigma-Aldrich, cat#146072) at 37 °C. Next, gelation was performed in ammonium persulfate (APS, Sigma-Aldrich, cat#A7460)/N,N,N′,N′-Tetramethyl ethylenediamine (Temed, Sigma-Aldrich, cat#110-18-9)/Monomer solution (23% Sodium Acrylate (SA, Sigma-Aldrich, cat#408220); 10% AA; 0.1% N,N′-Methylenebisacrylamide (BIS-AA, Sigma-Aldrich, cat#M7279) in PBS) for 1 h at 37 °C. Sample denaturation was performed for 90 min at 95 °C. Gels were incubated in bulk ddH2O at room temperature overnight for complete expansion. The following day, gel samples were washed in PBS twice for 30 min each to remove excess of ddH2O. Gels were then cut into square pieces (~1 cm × 1 cm), incubated with primary antibodies at 37 °C for 3 h, and washed with 0.1% PBS-Tween (PBS-T) 3 times for 10 min each.

Incubation with the secondary antibodies was performed for 3 h at 37 °C followed by 3 times washes with 0.1% PBS-T 10 min each. In some conditions if needed, gels were additionally stained by NHS-ester (Merck, cat#08741) diluted at 10 μg/mL in PBS for 90 min at room temperature. After the final staining step, gels were then washed with 0.1% PBS-T 3 times for 15 min each and expanded overnight by incubating in bulk ddH2O at room temperature. After the second round of expansion, gels were cut into square pieces (~0.5 cm × 0.5 cm) and mounted by a coverslip in a fixed position for image acquiring.

### TurboID-based proximity labeling coupled with U-ExM (PL-U-ExM)
We combined TurboID-based proximity labeling (PL) of bulk proteins with U-ExM to investigate fine structures of ookinete apical complex. Mature ookinetes expressing the APR-targeting biotinylizer in the in vitro 10 mL culture were incubated with 50 μM biotin (final concentration) for additional 3 h at 22 °C. Ookinetes were washed three times with PBS to remove unused biotin, and biotin-labeled ookinetes were collected and purified. The purity of the ookinetes was examined by hemocytometer analysis. Ookinetes with high purity (>80%) were separated into $3 \times 10^5$ per tube containing 500 μL of PBS for one U-ExM experiment. Thereafter the U-ExM steps were performed as mentioned above. For visualizing the biotin-labeled structure, Alexa Fluor 488-conjugated streptavidin (SA-488, 1:1000) were used after the secondary antibodies staining procedure in the U-ExM.

### Preparation of GMPCPP-stabilized microtubule seeds
Double-cycled GMPCPP microtubule (MT) seeds were made as described previously[50]. 8.25 μL tubulin reaction mixture in MRB80 buffer (80 mM Pipes, 1 mM EGTA, and 4 mM MgCl2, pH 6.8), which contained 14 μM unlabeled porcine brain tubulin (Cytoskeleton, cat#T238P-C), 3.6 μM biotin-tubulin (Cytoskeleton, cat#T333P), 2.4 μM rhodamine-tubulin (Cytoskeleton, cat#TL590M), and 1 mM GMPCPP (Jena Biosciences, cat#NU-405L), was incubated at 37 °C for 30 min. MTs were then pelleted by centrifugation in an Airfuge (Beckman) at 28 psi (pounds per square inch) for 5 min. The supernatant was carefully removed, and the pellet was resuspended in 6 μL MRB80 buffer and depolymerized on ice for 20 min. Subsequently, a second round of polymerization was performed at 37 °C in the presence of freshly supplemented 1 mM GMPCPP. The MT seeds were then pelleted as described above and resuspended in 50 μL MRB80 buffer containing 10% glycerol, snap frozen, and stored at −80 °C.

### TIRF assays, image acquisition, and data processing
Flow chambers for assays were made of plasma-cleaned glass coverslips and microscope slides. The flow chambers were sequentially incubated with 0.2 mg/mL Poly-L-Lysine-polyethylene glycol (PLL-PEG)-biotin (Susos AG) and 1 mg/mL neutravidin (Invitrogen, cat#31000) in MRB80 buffer. GMPCPP seeds were then attached to coverslip via biotin-neutravidin links, followed by blocking with 1 mg/mL κ-casein. The reaction mixture, which consisted of purified protein and MRB80 buffer containing 20 μM porcine brain tubulin, 0.5 μM rhodamine-tubulin, 1 mM GTP, 0.2 mg/mL κ-casein, 0.1% methylcellulose, and oxygen scavenger mix (50 mM glucose, 400 μg/mL glucose oxidase, 200 μg/mL catalase, and 4 mM DTT), was centrifuged with Airfuge for 5 min at 28 psi and added to the chamber. The flow chamber was sealed with vacuum grease and imaged immediately at 30 °C using a TIRF microscope. The imaging interval was 2 s unless otherwise stated. To generate curved MTs, GMPCPP seeds were first elongated in the presence of 20 μM tubulin for 5 min. Subsequently, the reaction mixture containing 30 nM purified APR2 and 10 μM tubulin was quickly flowed into the chamber with preassembled dynamic MTs. MT buckling appeared frequently because of the mechanical strain cause by the solution exchange.

TIRF microscopy was performed on Nikon Eclipse Ti2-E with the optimal focus with the Nikon CFI Apo TIRF 100 × 1.49 NA oil objective, Prime 95B camera (Photometrics), SOLE laser engine (four lasers: 405 nm, 488 nm, 561 nm, and 638 nm; Omicron) and controlled by NIS-Elements software (Nikon). Images were magnified with a 1.5 × intermediate lens on Ti2-E before projected onto the camera. The resulting pixel size is 73.3 nm/pixel. Stage top incubator INUBG2E-ZILCS (Tokai Hit) was used to keep cells at 37 °C or in vitro samples at 30 °C. Imaging medium (DMEM/F12 supplemented with 10% FBS, 100 U/mL penicillin and 100 μg/mL streptomycin) was prewarmed in a water bath at 37 °C. Optosplit III beamsplitter (Cairn Research Ltd.) was used for simultaneous imaging of green and red fluorescence. Stream acquisition was used for simultaneous imaging of green and red fluorescence in vivo. Microtubule dynamics were measured by producing kymographs using the Multi Kymograph function of the Fiji image analysis software and manually fitting lines to growth and shrinkage events. Microtubule growth and shrinkage velocities were calculated from the slopes of the fitted lines. Microtubule catastrophe frequencies were calculated as the inverse of the mean of microtubule lifetimes. Microtubule rescue frequencies were calculated as the inverse of the mean duration of depolymerization events; events without a rescue were assigned a value of 0.

## TurboID-based proximity-labeling and Pull-down

Purified ookinetes expressing the biotin ligase TurboID were incubated with 50 μM biotin (Sigma-Aldrich, cat#B4639) at 22 °C for 3 h and washed three times with PBS to remove the unused biotin. The ookinetes were collected and stored at −80 °C. For pull-down of biotinylated proteins, $1.0 \times 10^8$ ookinetes (one biological replicate) were lysed in RIPA buffer (50 mM Tris-HCl pH 7.4, 150 mM NaCl, 1% NP40, 0.1% SDS, 1% sodium deoxycholate, 1% TritonX-100, 1 mM EDTA) containing protease inhibitor cocktail and PMSF. 5 mg of total proteins from cell lysates (one biological replicate) were subjected to a 15 mL centrifuge tube containing 100 μL of streptavidin sepharose (Thermal Scientific, cat#SA10004). After incubation overnight at 4 °C, streptavidin beads were washed with the following procedures: twice with RIPA lysis buffer, once with 2 M urea in 10 mM Tris-HCl (pH 8.0), and twice with 50 mM Tris-HCl (pH 8.5). The washed beads containing biotinylated proteins were re-suspended in 200 μL 50 mM Tris-HCl (pH 8.5) for further digestion.

## Protein digestion and peptide desalting

The enriched biotinylated proteins were digested on-bead by rolling with 1 μg of trypsin for 16 h at 37 °C. For digested peptide samples, StageTips packed with SDB-RPS (2241, 3 M) material (made in-house) were used for desalting. About 1% trifluoroacetic acid (TFA, Sigma-Aldrich, cat#T6508) was added into the reactions to stop digestion. The SDB-RPS StageTips were conditioned with 100 μL 100% acetonitrile (ACN) (Sigma-Aldrich, cat#3485). The peptides were loaded into StageTips followed by centrifugation at 4000 g for 5 min. StageTips were washed twice with 100 μL 1% TFA/isopropyl alcohol (Sigma-Aldrich, cat#I9030), and then washed with 100 μL 0.2% TFA. Peptides were eluted with 80% ACN solution in 5% ammonia water. All eluted materials were collected in glass vials (CNW Technologies, cat#A3511040) and dried at 45 °C using a SpeedVac centrifuge (Eppendorf Concentrator Plus, cat#5305).

## Mass spectrometry

Digested peptides were dissolved in 0.1% formic acid (Sigma-Aldrich, cat#06440) containing independent retention time (iRT) peptides and analyzed by Sequential Window Acquisition of All Theoretical Mass Spectra (SWATH-MS) on TripleTOF 5600. For SWATH-MS, an MS1 scan records a 350 to 1250 m/z range for 250 ms, and a 100 to 1800 m/z range was recorded for 33.3 ms in the high-sensitivity mode MS2 scan. One MS1 scan was followed by 100 MS2 scans, which covered a precursor m/z range from 400 to 1200. SWATH-MS wiff files were converted to centroid mzXML files using MSConvert (version 3.0.19311), and were then subjected to DIA-Umpire software (version 2.1.6) for analysis. Signal-extraction module of DIA-Umpire was used to generate pseudo-DDA mgf files. These mgf files were converted to mzML files, which are subjected to database search using MSFragger (version 2.3) through the FragPipe interface (https://fragpipe.nesvilab.org/). The search parameters were set as followed: precursor mono-isotopic mass tolerance '50 ppm' fragment mass tolerance '0.1 Da', modification '57.021464@C', potential modification mass '15.994915@M', cleavage 'semi' and maximum missed cleavage sites '1'. PeptideProphet, ProteinProphet and FDR filtering were performed by Philosopher software (version 3.2.2) (https://github.com/Nesvilab/philosopher) through the FragPipe interface (https://fragpipe.nesvilab.org/)[51]. The pep.xml search results were validated and scored using PeptideProphet followed by analysis with ProteinProphet. The precursor ions and proteins were filtered at 1% FDR. The spectral library was generated by using EasyPQP tool (version 0.1.12) which is integrated in the FragPipe software. SWATH-MS files were converted to profile mzXML files. The spectral library based targeted analysis of SWATH-MS was performed using the QuantPipe tool based on the Open SWATH-PyProphet-Tric workflow[52,53]. The results were filtered at 1% global protein FDR. Statistical analysis by Perseus software (version 1.6.10.43) were performed as previously reported[54]. Parasite protein intensities were imported into Perseus. Protein abundances were normalized with total intensities of all proteins per run and then log2 transformed. The Pearson correlation analysis, hierarchical clustering, and volcano plots were performed with default settings.

## Proximity ligation assay (PLA)

PLA assay was performed to detecte in situ protein interaction using a commercial kit (Sigma-Aldrich, cat#DUO92008, DUO92001, DUO92005, and DUO82049). Ookinetes were fixed with 4% PFA for 30 min, permeabilized with 0.1% Triton X-100 for 10 min, and blocked with a blocking solution overnight at 4 °C. The primary antibodies were diluted in the Duolink Antibody Diluent and incubated with ookinetes in a humidity chamber overnight at 4 °C. The primary antibodies were removed, and parasites were rinsed twice with wash buffer A. The PLUS and MINUS PLA probes were diluted in Duolink Antibody Diluent and incubated in a humidity chamber for 1 h at 37 °C. Next, ookinetes were rinsed twice with wash buffer A and incubated with the ligation solution for 30 min at 37 °C. Followed by twice rinses with wash buffer A, ookinetes were incubated with the amplification solution for 100 min at 37 °C in the dark. After rinsing twice with 1 × wash buffer B and once with 0.01 × wash buffer B, ookinetes were stained with Hoechst 33342 and washed twice with PBS. Images were captured and processed using identical settings on a Zeiss LSM 880 confocal microscope.

## Yeast two-hybrid assay

Yeast two-hybrid assay (Y2H) was performed using the MATCHMAKER GAL4 Two-Hybrid System (Clontech) according to the manufacturer's instructions. pGBKT7 (express any protein as a GAL4 DNA-BD fusion) and pGADT7 (express any protein as a GAL4 AD fusion) were used as cloning vectors in our Y2H experiments. pGADT7-RecT (express SV40 large antigen as a GAL4 AD fusion) co-transfection with pGBKT7-53 (express p53 as a GAL4 DNA-BD fusion) was used as a positive control and pGBKT7-Lam (express Lamin C as a GAL4 DNA-BD fusion) was used as negative control. In the APR2 interacting protein(s) screening, two split fragments F1 (1–1800 bp) and F2 (1801–4185 bp) from *apr2* coding sequence (1–4185 bp) were individually cloned into the pGBKT7 vector as the baits, while cDNAs of the top 10 APR2 PL protein hits were cloned into the pGADT7 vector as the preys. For detecting the interaction between APRp2 and APRp4, cDNAs of APRp2 and

APRp4 were cloned into pGADT7 and pGBKT7 respectively. Pairs of constructs were transformed into the yeast strain AH109. Transformants were selected on DDO medium (SD-2: -Leu/-Trp) at 30 °C for 3 d, then the selection of interactions was made on QDO medium (SD-4: -Ade/-His/-Leu/-Trp) at 30 °C for 3 d. The experiments were repeated twice independently.

## Bioinformatic searches and tools

The genomic sequences of target genes were downloaded from PlasmoDB database (https://plasmodb.org/plasmo/app/)[55]. The phylogenetic tree and protein amino acid sequence alignment was generated using MEGA5.0[56]. The genomic sequences of target genes were downloaded from PlasmoDB. The sgRNA of target genes were searched using database EuPaGDT (http://grna.ctegd.uga.edu/)[57]. The codon usage was optimized using JCat (http://www.prodoric.de/JCat)[58] The cluster heatmap comparing two data collections from Tb-APR2 and Tb-ARA1 was generated using the web-based Morpheus tool from the Broad Institute (https://software.broadinstitute.org/morpheus/)[59]. Amino acid charge analysis was performed by CIDER (http://pappulab.wustl.edu/CIDER/analysis/)[60].

## Quantification and statistical analysis

For quantification of protein relative expression in immunoblot, protein band intensity was quantified using by Fiji software[61]. For quantification of protein-protein relative co-localizaiton in immunofluorescent images, apical end of each measured ookinete was set as an ROI (region of interest) to calculate the Pearson correlation coefficient by Fiji sofaware[61]. Distance from plasma membrane to subpellicular microtubules and structure size parameters (volume and area) of Z-stack images were quantified by Imaris X64 9.2.0. For quantification of protein expression in IFA, images were acquired under identical parameters. Fluorescent signal intensities and areas were quantified using ZEN Microscopy Software from ZEISS. More than 50 cells were randomly chosen in each group. All graph-making and statistical analysis were performed using GraphPad Prism 8.0. Data collected as raw values were shown as means ± SEM, means ± SD or means only if not otherwise stated. Details of statistical methods were reported in the figure legends. Two-sided $t$-test or Mann–Whitney test were used to compare differences between control and experimental groups. Statistical significance was shown as $*p < 0.05$, $**p < 0.01$, $***p < 0.001$, $****p < 0.0001$, ns, not significant. n represents the sample volume in each group or the number of biological replicates.

## Reporting summary

Further information on research design is available in the Nature Portfolio Reporting Summary linked to this article.

# Data availability

Mass spectrometry proteomics data have been deposited to the ProteomeXchange Consortium (http://proteomecentral.proteomexchange.org) via the iProX partner repository[62]. The dataset identifier is PXD038209. All other relevant data in this study are submitted as supplementary source files. Source data are provided with this paper.

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

## Acknowledgements

We thank Dr. Xinzhuan Su (NIH/NIAID) and Dr. Ziyin Li (University of Texas Health Science Center at Houston) for their comments on this manuscript. This work was supported by the National Natural Science Foundation of China (32270503, 32200554, 32170427, 31970387, 31872214), the Natural Science Foundation of Fujian Province (2021J01028), and the 111 Project sponsored by the State Bureau of Foreign Experts and Ministry of Education of China (BP2018017).

## Author contributions

P.Q. and X.W. generated the modified parasites, conducted the phenotype analysis, IFA assay, image analysis, mosquito experiments, and biochemical experiments. C.G. performed the in vitro and in vivo MT-binding experiments, X.F. and M.C. generated the modified parasites, C.Z. conducted the MS, Y.L. and Y.C. performed the yeast two hybrid assay, L.Y. performed the electronic microscopy experiments, J.Y., H.C., and K.J. supervised the work, P.Q., X.W., H.C., and J.Y. analyzed the data, and J.Y. wrote the manuscript.

## Competing interests

The authors declare no competing interests.
