## [Peer Review File · Nature Communications]

Apical anchorage and stabilization of subpellicular microtubules by apical polar ring ensures *Plasmodium* ookinete infection in mosquitoEditorial Note: Parts of this Peer Review File have been redacted as indicated to remove third-party material where no permission to publish could be obtained.

REVIEWER COMMENTS

Reviewer #1 (Remarks to the Author):

Summary:

The structure of subpellicular microtubules nucleated from the apical polar ring (APR) is essential for parasite morphogenesis, gliding and invasion. Although the APR is widely accepted as the microtubule organizing center (MTOC) for nucleating and anchoring subpellicular microtubules in Plasmodium zoites, the exact function of APR proteins is yet to be verified. The manuscript by Qian et al. studies the function of APR protein APR2 to reveal that APR2 is essential for the APR structural integrity and impacts ookinete morphogenesis, gliding motility and mosquito transmission in Plasmodium yoelii. The authors applied expansion microscopy and electron microscopy to reveal the structure of APR, and the relationship of APR with subpellicular microtubules and inner membrane complex in ookinetes. This study demonstrates that APR2 and its APR-residing partners APRp2 and APRp4 are essential for maintaining APR structural integrity and functional fulfillment. Furthermore, this study also indicates the function of APR2 is specific in ookinetes, but not critical in merozoites, although it is a good merozoite and sporozoite marker.

I enjoyed reviewing this large and impressive work, which revealed the structure of APR and the function of APR2 in Plasmodium ookinetes. However, I have some comments and suggestions for the authors of this manuscript prior to its publication.

Major comments:

1. To verify that the APR-2 N-terminus has MT-binding and MT-stabilization properties, the authors performed experiments expressing the Plasmodium APR2-N fragment protein in vivo and in vitro in mammalian cells (Figure 5). The APR2-N truncated fragment episomes data indicated the APR2-N with MT-binding ability can target APR in the WT ookinetes (Supplementary Figure 4D). The deletion of the APR2 N-terminus caused the reduction of fluorescent signals at APR in APR2- Δ N ookinetes (Figure 5 A-C). Did the authors try the expansion microscopy in APR2- Δ N ookinetes to support that the deletion of the APR2 N-terminus caused the APR2 C-terminus to dissociate from subpellicular microtubules?

2. To validate that the APR2 C-terminus can localize APR, the authors generated the APR2- Δ N, which still showed APR localization after the deletion of the APR2 N-terminus (Figure 5 A-C). They also further verified this result by expressing APR2-M and APR2-C HA-fused episomes in the WT and Δ apr2 ookinetes (Figure 5D-F). The authors also provided the expression of APR2-N episome in WT ookinetes (Supplementary Figure 4D). Did the authors also have data of APR2-N episome expression in the Δ apr2 ookinetes? I would suggest the authors include that data in the Figure 5E.

Minor comments:

Introduction:

Line 56: "60 SPMTs in ookinetes" was only reported in reference 4 and was not found in references 8-12 as the author cited.

Line 57: The author cited two references to indicate the number of SPMTs is 3-5 in merozoites. But in *P. falciparum* merozoites the number of SPMTs is 1-2 in reference 14, and 2-3 in reference 15. A recent study that applied expansion microscopy found typically 2-4 SPMTs in *P. falciparum* merozoites (Liffner and Absalon 2021).

Results:

Line 170: The authors forgot to describe the co-localization of APR2 with microneme proteins (chitinase and CTRP, Figure 2C) in the manuscript.

Line 260: APR2-N "MT-stabilizing activity increases in a dose-dependent manner". However, microtubule rescue values were reduced with the increase in APR2-N concentrations (10 to 20 nM) in Figure 4G.

Figures:

Figure 1K and Figure 9E: How to measure the θ in the wide type, $\Delta apr2$, $\Delta aprp2$ and $\Delta aprp4$ ookinetes, and how to select the point in the nucleus to generate the lines of nucleus-apical and nucleus-basal? Is the centroid or center of mass? It would be better to clarify it in the figure legend or the method.

Figure 2C: The authors forgot to describe the images are live cells or immunofluorescence assay in the figure legend.

Figure 4G and 4I: What is the microtubule growth result looks like? EB1 induces microtubule catastrophes and increases growth velocity in vitro as manuscript reference 31 described. Furthermore, the EB1 was reported can stabilize the microtubules in *Toxoplasma gondii* (Chen, Kelly et al. 2015). The Plasmodium EB1 function may be more like TgEB1 to stabilize microtubules rather than facilitate microtubules catastrophe. It would be better if the authors could point out the difference in EB1 between apicomplexan cells and mammalian cells in discussion.

Figure 6B: It would be better for the authors to point out the location of micronemes.

Figure 7E: From the TEM data, it does not look like all the SPMTs lost the association with IMC, it would be better for "SPMTs lost association with IMC in the $\Delta apr2$ ookinete" to change as "x% SPMTs lost association with IMC in the $\Delta apr2$ ookinete" in the figure legend.

Figure 9B: "BF/Hoechst" correspondent images are 3 channels merged. The authors should change the "BF/Hoechst" to Merge.

Supplementary Figure 3C: It would be better for the authors to provide zoom-in images for the photo-bleached area.

Supplementary Figure 6B: "BF/Hoechst" correspondent images are 4 channels merged. The author should change the "BF/Hoechst" to Merge.

Discussion:

Line 551: has the same issue as line 56.

Line 552: has the same issue as line 57.

Line 618: "APR2 binds to and stabilizes MTs in both the heterologous mammalian expression and the in vivo assays", the data did not verify that APR2 can stabilize MTs in vivo. It would be good to point out in the discussion of future work that microtubule depolymerization agents, trifluralin or colchicine, could be used to verify APR2 can stabilize SPMTs in WT and $\Delta apr2$ ookinetes

Chen, C. T., M. Kelly, J. Leon, B. Nwagbara, P. Ebbert, D. J. Ferguson, L. A. Lowery, N. Morrisette and M. J. Gubbels (2015). "Compartmentalized *Toxoplasma* EB1 bundles spindle microtubules to secure accurate chromosome segregation." *Mol Biol Cell* 26(25): 4562-4576.

Liffner, B. and S. Absalon (2021). "Expansion Microscopy Reveals *Plasmodium falciparum* Blood-Stage Parasites Undergo Anaphase with A Chromatin Bridge in the Absence of Mini-Chromosome Maintenance Complex Binding Protein." *Microorganisms* 9(11).

Reviewer #2 (Remarks to the Author):

Qian et al. describe a comprehensive and impressive study of the apical polar ring (APR) in ookinetes of the rodent-infecting malaria parasite, *Plasmodium yoelii*. The authors use a sophisticated and deep toolbox comprised of cell biology, molecular genetics, biochemistry, microscopy and proteomics to unveil novel important phenotypes and proteins that control the development of the APR for ookinete shape, motility, and infectivity, all of which are intrinsically linked and required for malaria parasite transmission. The study is generally well constructed,

controlled, executed and concluded. The manuscript describes a very large body of work with conclusions that are sound. It is generally written well and has logical flow, although some revision of grammar and experimental studies is recommended (see below).

I did not fully understand the experiments shown in Figure 4D-E as this is outside my expertise, so I could not comment on the results or conclusions thereof. For example, it was not defined what catastrophe or rescue means and I could not observe a dose-dependent response for these in Panel G and I.

Major Points

There is no genotyping data provided that validates the molecular genetics approaches employed in this study. Please provide a summary file containing confirmation of all genotypes for each transgenic line produced in the study (tags, gene disruptions, etc.) by PCR, Southern blot, sequencing, or other preferred method(s).

To accurately claim in Figure 1 that APR2 regulates ookinete shape and gliding, direct data for the APR2n mutant versus the complemented line is required (eg. Figure panels K, L; see lines 119, 147).

Please check that every figure panel is referenced in the text. There are examples of some that currently are not (eg. FigS1E).

In Figure S2C, female is marked by absence of tubulin, but this is not a reliable marker, as females do also label with this. Is there another female-specific marker that can be used? For example, g377 is a female marker.

This study describes the function of conserved proteins in Plasmodium spp using *P. yoelii*. It is important to also discuss these findings in the context of human disease caused by pathogens such as *P. falciparum*, *P. vivax*, *P. malariae* etc. Please comment on the conservation of APR2, ARA1, APRp1-9 in the human pathogenic species.

Minor Points

Throughout the manuscript. Check grammar for correct use of plural language, (eg. line 109, oocysts) sentence structure (eg. line 100, 'the' mosquito; line 156, apical 'end' or 'tip'), etc. Also, use of the term 'dot' or 'circle' is not very scientifically descriptive nomenclature for microscopy images. Line 159 merozoites and sporozoites.

Line 101, I suggest "less cell bending" not "no cell bending", as the angle is not 180, indicating there is still some slight bending.

Line 183, I suggest inserting some words so the sentence reads "...many short HA-labelled spines..." or similar.

Line 188, 193 can you please fix the sentence (eg. 'apical tip' or 'apical end') not just saying 'apical'?

Line 191, the apical co-localization seems partial, and I recommend using a Pearson's correlation.

Line 236, These results do not prove strong binding behavior as stated, but co-localization. An immunoprecipitation or proximity ligation (PL) microscopy assay is required to conclude that they interact.

Line 269, It would be clearer just to state MRC5 cells.

Line 326, 329 need quantification like lines 318 and 320.

Line 328 suggests not indicates.

Line 345, sentence should state 'after genetic disruption' (not after loss).

Line 348, signals? Do the authors mean EM density? This requires clarification and the data should be quantified from numerous observations.

Line 369, 'severely destroyed' is too strong a conclusion in my opinion. See the merge in Figure 6F apr2mutant, they are still there but fainter (as per the reduced signal area and volume).

Line 391, the terminology "almost destroyed" is not the best scientific nomenclature, can an alternative be used?

Line 410, the number of micronemes in the posterior versus anterior halves of the ookinetes should be quantified to support this claim.

Line 487, for completion, do the authors have data for ARA1 mutant phenotypes for ookinete formation, crescent shape, gliding speed and oocyst numbers?

For all blots, please indicate if possible, how many ookinetes were loaded into each gel lane for reproducibility of Western blots (eg. Figure S2, etc).

In Figure 1, while it is shown that mature (presumably stage V) ookinetes are produced less, it is not clear at what stage (stage I, II, III or IV) the ookinetes arrest at. Can the authors quantify stages and show when the difference(s) occurs (eg Figure 1N)? See Figure 2A for example of the different stages.

Figure S1. In panel D, a zygote was shown for 17XNL but a retort for APR2 mutant. This makes the figure more confusing (I thought the micrographs were to follow the schematic above, but this is not correct and it took some decoding). Please show similar representatives of the same stages in the micrographs. To assist with the confusion, could the schematic be placed into supplementary, or separated further from the microscopy? In panel E, it is best to ANOVA not t-test here and throughout the study for multiple comparisons.

Figure 2. In panel B, the Brightfield channel is too dark. In panel C, please indicate which channel is HA and which is Myc.

Figure S3A. It is not clear in the cross section which parts are SPMT until the very high zoom. Can this be made clearer with a different zoomed inset or color?

In Figure 4C, Did FL APR2(1-1394)-GFP also bind Tubulin by IP?

In Figure S4A, B the data are not proof of MT binding (line 249), but of co-localization. Would require an immunoprecipitation or PL microscopy assay to prove they interact.

In Figure 5B, the APR2-dN ookinete looked straighter than the apr2::gfp control ookinete. Was this measured? It likely indicates the region of APR2 that contributes to the phenotype. Can this be quantified from other images and included?

Figure 6, panels C and D require arrows to show the details that are altered in dAPR2.

Figure 7G, use of densitometry for the Western blot would be useful. How many times was this repeated? It would be helpful to briefly define the treatments in the figure legend.

Figure S7C, the data circles in the black histogram bars are camouflaged, can these be another color to see them?

Figure S8G, can the colloidal gold particles be quantification be provided to support the claim (line 515)?

Reviewer #3 (Remarks to the Author):

This is an elegant study of the composition, disposition and function of the apical polar ring (APR) of *Plasmodium* ookinetes. The establishment of a motile and invasive polarized cell is a critical event in zygote to ookinete differentiation for this group of parasites to allow transmission through the mosquito vector, as well as being essential for other invasive stages of the life cycle.

This work uses genetic modification, ultrastructural studies and proteomics to make a considerable contribution to our understanding of APR structure.

The work has been cleanly executed, the data are clear and very nicely presented, and the results represent a significant contribution to the area.

I have little to criticize in the study and believe it will be of considerable interest to, for example, cell biologists, particularly those with an interest in apicomplexan parasites.

Minor points:

Line 51; my understanding is that the pellicle includes the parasite plasma membrane.

Throughout the text and supplementary figures (including figure axes); the English needs to be checked for typos and grammar, and corrected as appropriate.

Reviewer #4 (Remarks to the Author):

The study by Qian et al. characterizes a defining structure of the apicomplexan phylum, the apical polar ring (APR), with unprecedented detail in the ookinete. They provide an extensive phenotypic analysis of a critical component of this structure, APR2, and expand the APR proteome by some novel component. This study presents an excellent body of work including very attractive super-resolution imaging that unpacks the role of APR2 in a series of logical steps. Their conclusions are supported by the remarkably large and well-controlled dataset, which is even complemented by analyses in mammalian cell model, and only requires minor improvements. The study provides very new insights into the organization critical for multiple life cycle stages of the parasite with relevance for other apicomplexan species. Taken together the authors provide an excellent cell biological framework for any further study of the APR as an atypical microtubule organizing center. Hence, I have no reservations to recommend this manuscript for publication in your journal after the following comments have been properly addressed.

Major comments

Fig. 1A: The integration PCR results for the *apr2* mutant strains, which is also mentioned in methods, should be shown (even though the phenotypes look convincing).

Fig. 3B: Negative controls with only one antibody for the PLA in need to be shown (as well as for Fig. 9B).

Fig. 4E: Analyses of changes of microtubule dynamics shown in Fig. 4E are very enlightening. Did something prevent the orders from using the full length APR2 (seen in Fig. 4A) in this experiment? If possible it would strengthen the conclusions by the authors about the function of APR2.

If possible, please show colocalization analysis of ARA1 with APR2 to learn more about their relative positioning.

Fig. 7B: Add total tubulin intensity to highlight microtubule stability difference.

Fig. S7: Add IFA images (ideally ExM) of micronemes in *apr2* KO to confirm EM images and claims about microneme phenotype.

Fig. 9A: The panel supposedly showing the Y2H results for interaction with APR2-F1 fragment

seems completely dark even though at least some background should be visible. Please add panel or adjust contrast.

Minor comments

Line 17: Find another formulation than "possessing".

Line 188: complete title.

Line 216: Replace the term "fluidity" with an alternative, like turnover.

The authors could comment a bit more about the inner and outer APR highlighted by Ferreira et al. in the context of the likely relative position of APR2 and associated proteins.

Response to Reviewer Comments on the manuscript [NCOMMS-22-27179A]:

Reviewer #1

The structure of subpellicular microtubules nucleated from the apical polar ring (APR) is essential for parasite morphogenesis, gliding and invasion. Although the APR is widely accepted as the microtubule organizing center (MTOC) for nucleating and anchoring subpellicular microtubules in *Plasmodium* zoites, the exact function of APR proteins is yet to be verified. The manuscript by Qian et al. studies the function of APR protein APR2 to reveal that APR2 is essential for the APR structural integrity and impacts ookinete morphogenesis, gliding motility and mosquito transmission in *Plasmodium yoelii*. The authors applied expansion microscopy and electron microscopy to reveal the structure of APR, and the relationship of APR with subpellicular microtubules and inner membrane complex in ookinetes. This study demonstrates that APR2 and its APR-residing partners APRp2 and APRp4 are essential for maintaining APR structural integrity and functional fulfillment. Furthermore, this study also indicates the function of APR2 is specific in ookinetes, but not critical in merozoites, although it is a good merozoite and sporozoite marker.

I enjoyed reviewing this large and impressive work, which revealed the structure of APR and the function of APR2 in *Plasmodium* ookinetes. However, I have some comments and suggestions for the authors of this manuscript prior to its publication.

Major comments:

1. To verify that the APR-2 N-terminus has MT-binding and MT-stabilization properties, the authors performed experiments expressing the *Plasmodium* APR2-N fragment protein in vivo and in vitro in mammalian cells (Figure 5). The APR2-N truncated fragment episomes data indicated the APR2-N with MT-binding ability can target APR in the WT ookinetes (Supplementary Figure 4D). The deletion of the APR2 N-terminus caused the reduction of fluorescent signals at APR in APR2- Δ N ookinetes (Figure 5 A-C). Did the authors try the expansion microscopy in APR2- Δ N ookinetes to support that the deletion of the APR2 N-terminus caused the APR2 C-terminus to dissociate from subpellicular microtubules?

Response:

In this study, we first established that APR2-N fragment has MT-binding and APR-targeting activity (see Fig. 4 and Supplementary Fig. 4). In the parallel, the APR2-C did not show MT-binding activity from both in vitro MT-binding assay and mammalian cell assay (see Fig. 4), but it could localize at the impaired APR structure in the APR2-null or APR2-N null ookinetes (see Fig. 5a, b, d, e, g and i). These results suggested that APR2-C targets to APR by interacting with other APR-localizing proteins.

Consistent with this speculation, in the last part of this study we identified two APR proteins APRp2 and APRp4. APRp2 and APRp4 directly interacted with APR2-C fragment, but not with APR2-N fragment from the yeast-two hybrid assay (see Fig. 9a). The interaction of APR2-C with APRp2 and APRp4 may explain the APR-targeting for APR2-C in the APR2-null or APR2-N null ookinetes.

After genetic disruption of APR2, the APR structure was severely impaired and most SPMTs lost apical anchorage with APR (see the U-ExM image in Fig. 7d). In this situation, it is technically difficult to distinguish the association or dissociation of APR2-C fragment from the SPMTs. Therefore, we didn't perform the U-ExM for analyzing the localization of APR2-C fragment in the APR2-null or APR2-N null ookinetes.

2. To validate that the APR2 C-terminus can localize APR, the authors generated the APR2- Δ N, which still showed APR localization after the deletion of the APR2 N-terminus (Figure 5 A-C). They also further verified this result by expressing APR2-M and APR2-C HA-fused episomes in the WT and Δ apr2 ookinetes (Figure 5D-F). The authors also provided the expression of APR2-N episome in WT ookinetes (Supplementary Figure 4D). Did the authors also have data of APR2-N episome expression in the Δ apr2 ookinetes? I would suggest the authors include that data in the Figure 5E.

Response:

In our initial experiments, we did test the APR2-N episome expression in the Δ apr2 ookinetes. The results showed that episomal APR2-N was still localized at apical in the Δ apr2 ookinetes (see the below picture, not included in the revised manuscript).

In the manuscript we have concluded that APR2-N has MT-binding and APR targeting activity (Fig. 4 and Supplementary Fig. 4). In the Fig. 5, we mainly attempted to analyze APR2-M and APR2-C for the APR-targeting activity. This is the logic we presented. Therefore, it is better to not include the results of the APR2-N episomal expression in the $\Delta apr2$ ookinetes.

Minor comments:

Introduction:

Line 56: "60 SPMTs in ookinetes" was only reported in reference 4 and was not found in references 8-12 as the author cited.

Response: In other previous studies from the references 8-12 (PMID: 11875126, 21118920, 13897014, 11562165, 5794450), the ookinete SPMTs from different *Plasmodium* species were observed or summarized. Therefore we cited these references in our manuscript. We listed the related texts and figures from these references in the below.

Reference-8 (PMID: 11875126):

[REDACTED]

Reference-9 (PMID: 21118920):

[REDACTED]

Reference-10 (PMID: 13897014):

[REDACTED]

Reference-11 (PMID: 11562165):

[REDACTED]

Reference-12 (PMID: 5794450):

[REDACTED]

Line 57: The author cited two references to indicate the number of SPMTs is 3-5 in merozoites. But in *P. falciparum* merozoites the number of SPMTs is 1-2 in reference 14, and 2-3 in reference 15. A recent study that applied expansion microscopy found typically 2-4 SPMTs in *P. falciparum* merozoites (Liffner and Absalon 2021).

Response:

Thank reviewer for pointing it out. We changed “3-5 SPMTs in merozoites” to “1-4 SPMTs in merozoites” and added the reference (Liffner and Absalon 2021, PMID: 34835432) in the revised manuscript.

Results:

Line 170: The authors forgot to describe the co-localization of APR2 with microneme proteins (chitinase and CTRP, Figure 2C) in the manuscript.

Response: Thank reviewer for pointing it out. We have added the description of those result in the revised manuscript (Line 169-170).

Line 260: APR2-N “MT-stabilizing activity increases in a dose-dependent manner”. However, microtubule rescue values were reduced with the increase in APR2-N concentrations (10 to 20 nM) in Figure 4G.

Response:

We calculated p value for the MT rescue values in Fig. 4g (see below). The results showed that MT rescue value from the three groups (10, 20, and 50 nM) were significantly increased compared to the control group (0 nM), but they did not show significant difference among the three groups (10, 20, and 50 nM).

In Fig. 4g, we did observe a clear dose-dependent decrease of both MT shrinkage and MT catastrophe with increasing concentration of APR2-N, indicating the MT-stabilization ability of APR2-N. Similarly, this dose-dependent effect of APR2-N was also detected for MT shrinkage in the presence of a well-known MT destabilizer (MT catastrophe-boosting factor) human EB1 (Fig. 4i). It is reasonable that when there is less MT catastrophe, there is less need for MT rescue in the *in vitro* MT assembly. Therefore, the MT rescue activity in our experiment conditions may reach a plateau with the frequency range between 4 and 6 (see the red dash lines in the picture below) in both Fig. 4g and Fig. 4i.

Figures:

Figure 1K and Figure 9E: How to measure the θ in the wide type, Δapr2 , Δaprp2 and Δaprp4 ookinetes, and how to select the point in the nucleus to generate the lines of nucleus-apical and nucleus-basal? Is the centroid or center of mass? It would be better to clarify it in the figure legend or the method.

Response: In each ookinete measured, three points were selected: apical end point, nucleus center point, and basal end point. The nucleus center point was set as vertex of the angel (θ). We added these information in the revised legend of Fig. 1k and Fig. 9e.

Figure 2C: The authors forgot to describe the images are live cells or

immunofluorescence assay in the figure legend.

Response: Thank reviewer for pointing it out. We added the information in the revised legend for Fig. 2b and 2c.

Figure 4G and 4I: What is the microtubule growth result looks like? EB1 induces microtubule catastrophes and increases growth velocity *in vitro* as manuscript reference 31 described. Furthermore, the EB1 was reported can stabilize the microtubules in *Toxoplasma gondii* (Chen, Kelly et al. 2015). The *Plasmodium* EB1 function may be more like TgEB1 to stabilize microtubules rather than facilitate microtubules catastrophe. It would be better if the authors could point out the difference in EB1 between apicomplexan cells and mammalian cells in discussion.

Response:

The MT growth analysis for Fig. 4g and 4i have been carried out in original experiments. The pictures below showed the results (not included in the revised manuscript). In each result of these experiments, the MT growth did not show subtle difference among groups. Because of the space limit, these results were not included.

Using the *in vitro* MT-binding assay, we found that APR2-N had the MT-stabilizing activity. To further confirm this, we evaluated the MT-stabilizing activity of APR2-N in the presence of a MT-destabilizer human EB1 (100 nM), which has been reported to facilitate MT catastrophe *in vitro* (PMID: 24508171, 18059460). In the presence of human EB1, APR2-N still executed a protection effect on MT stabilization (Fig. 4h and i). These results further confirmed that APR2-N is able to stabilize MTs *in vitro*. In this study, the *Plasmodium* EB1 is not used. I agreed with the comments that EB1 proteins from apicomplexan and mammalian could be different in the MT-binding or MT-modulating properties. Therefore, we prefer not to include it in the discussion section.

Figure 6B: It would be better for the authors to point out the location of micronemes.

Response: Thanks for reviewer's suggestion. We have marked the micronemes with asterisks in the revised Fig. 6b.

Figure 7E: From the TEM data, it does not look like all the SPMTs lost the association with IMC, it would be better for “SPMTs lost association with IMC in the $\Delta apr2$ ookinete” to change as “x% SPMTs lost association with IMC in the $\Delta apr2$ ookinete” in the figure legend.

Response:

We agreed with the reviewer. We changed the “SPMTs lost association with IMC in the $\Delta apr2$ ookinete” to “approximately half of SPMTs lost association with IMC in the $\Delta apr2$ ookinete”.

Figure 9B: "BF/Hoechst" correspondent images are 3 channels merged. The authors should change the "BF/Hoechst" to Merge.

Response: Changed.

Figure S3C: It would be better for the authors to provide zoom-in images for the photo-bleached area.

Response: Thanks for reviewer's suggestion. We have added the zoom-in images in the revised Supplementary Fig. 3c.

Figure S6B: "BF/Hoechst" correspondent images are 4 channels merged. The author should change the "BF/Hoechst" to Merge.

Response: Changed.

Discussion:

Line 551: has the same issue as line 56.

Response: We have corrected it accordingly.

Line 552: has the same issue as line 57.

Response: We have corrected it accordingly.

Line 618: “APR2 binds to and stabilizes MTs in both the heterologous mammalian expression and the *in vivo* assays”, the data did not verify that APR2 can stabilize MTs *in vivo*. It would be good to point out in the discussion of future work that microtubule depolymerization agents, trifluralin or colchicine, could be used to verify APR2 can stabilize SPMTs in WT and $\Delta apr2$ ookinetes

Response:

We have changed the “APR2 binds to and stabilizes MTs in both the heterologous mammalian expression and the *in vivo* assays” to “APR2 binds to and stabilizes MTs in both the heterologous mammalian expression and the *in vitro* assays” (line 622-623). In addition, we thank reviewer’s suggestion and have added those insightful ideas in the discussion of the revised manuscript (line 623-625).

Reviewer #2 (Remarks to the Author):

Qian et al. describe a comprehensive and impressive study of the apical polar ring (APR) in ookinetes of the rodent-infecting malaria parasite, *Plasmodium yoelii*. The authors use a sophisticated and deep toolbox comprised of cell biology, molecular genetics, biochemistry, microscopy and proteomics to unveil novel important phenotypes and proteins that control the development of the APR for ookinete shape, motility, and infectivity, all of which are intrinsically linked and required for malaria parasite transmission. The study is generally well constructed, controlled, executed and concluded. The manuscript describes a very large body of work with conclusions that are sound. It is generally written well and has logical flow, although some revision of grammar and experimental studies is recommended (see below).

Response: We thank the reviewer for the encouraging comments on our work.

I did not fully understand the experiments shown in Figure 4D-E as this is outside my expertise, so I could not comment on the results or conclusions thereof. For example, it was not defined what catastrophe or rescue means and I could not observe a dose-dependent response for these in Panel G and I.

Response:

Detailed descriptions for quantifying the MT behavior including MT shrinkage, MT catastrophe, and MT rescue were provided in the Method section (Line 1040-1044). We attached one example figure (see below) from PMID: 35051355 for illustrating the in vitro MT dynamics parameters.

[REDACTED]

In Fig. 4g, we did observe a clear dose-dependent decrease of both MT shrinkage and MT catastrophe with increasing concentration of APR2-N, indicating the MT-stabilization ability of APR2-N. Similarly, this dose-dependent effect of APR2-N was also detected for MT shrinkage in the presence of a well-known MT destabilizer (MT catastrophe-boosting factor) human EB1 (Fig. 4i). It is reasonable that when there is

less MT catastrophe, there is less need for MT rescue in the in vitro MT assembly. Therefore, the MT rescue activity in our experiment conditions may reach a plateau with the frequency range between 4 and 6 (see the two red lines in the picture below) in both Fig. 4g and Fig. 4i.

Major Points

There is no genotyping data provided that validates the molecular genetics approaches employed in this study. Please provide a summary file containing confirmation of all genotypes for each transgenic line produced in the study (tags, gene disruptions, etc.) by PCR, Southern blot, sequencing, or other preferred method(s).

Response: We have added a new supplementary figure (Supplementary Fig. 11), which provided the PCR genotyping results for all the genetically modified parasite strains generated in this study.

To accurately claim in Figure 1 that APR2 regulates ookinete shape and gliding, direct data for the APR2n mutant versus the complemented line is required (eg. Figure panels K, L; see lines 119, 147).

Response: Thanks for the reviewer's suggestion. We have added the information in the revised Fig. 1k and i.

Please check that every figure panel is referenced in the text. There are examples of

some that currently are not (eg. FigS1E).

Response: Thank the reviewer for pointing it out. We added the reference for Supplementary Fig. 1e in the text (line 121) and checked the whole manuscript thoroughly.

In Figure S2C, female is marked by absence of tubulin, but this is not a reliable marker, as females do also label with this. Is there another female-specific marker that can be used? For example, g377 is a female marker.

Response:

Generally, male gametocytes are recognized by cytosolic abundant tubulin protein signal after antibody staining while female has only weak tubulin signal in the nucleus. In addition, the Hoechst 33342 signal in nuclear area is larger in size in male than that in female. We presented some extra images for male and female *apr2::6HA* gametocytes co-stained with tubulin and HA antibodies (see the below picture, not included in the revised manuscript).

G377 is a well-established marker for female gametocyte in *P. falciparum*. Unfortunately, we don't have the antiserum against PyG377 in the lab.

This study describes the function of conserved proteins in *Plasmodium* spp using *P. yoelii*. It is important to also discuss these findings in the context of human disease caused by pathogens such as *P. falciparum*, *P. vivax*, *P. malariae* etc. Please comment on the conservation of APR2, ARA1, APRp1-9 in the human pathogenic species.

Response:

Thanks for reviewer's suggestion. We analyzed the conservation of amino acid sequences for these APR proteins among *Plasmodium* species (including three human parasites and three rodent parasites). Except APRp3, the other 9 proteins (APR2, ARA1,

APRp1-2, and APRp4-9) have orthologs showing variable degree of homology among human and rodent malaria parasites. These results suggest the conserved mechanism for APR2-regulated APR stability and apical anchorage of SPMT in *Plasmodium* including the *P. falciparum*, *P. vivax*, *P. malariae*. We added these information in the new Supplementary Fig. 10 and in the revised discussion section (line 606-611).

Minor Points

Throughout the manuscript. Check grammar for correct use of plural language, (eg. line 109, oocysts) sentence structure (eg. line 100, ‘the’ mosquito; line 156, apical ‘end’ or ‘tip’), etc. Also, use of the term ‘dot’ or ‘circle’ is not very scientifically descriptive nomenclature for microscopy images. Line 159 merozoites and sporozoites.

Response: We changed the oocysts in line 103; changed to mosquito in line 104-105; change the apical end in line 151; deleted the dot or circle in line 154; changed to merozoites and sporozoites in line 155.

Line 101, I suggest “less cell bending” not “no cell bending”, as the angle is not 180, indicating there is still some slight bending.

Response: we changed the “no cell bending” to “less cell bending” (line 127).

Line 183, I suggest inserting some words so the sentence reads “...many short HA-labelled spines...” or similar.

Response: we changed the “many short spines” to “many short HA-labelled spines” (line 178-179).

Line 188, 193 can you please fix the sentence (eg. ‘apical tip’ or ‘apical end’) not just saying ‘apical’?

Response: we changed the subtitle “APR2 associates with SPMTs at apical” to “APR2 associates with apical SPMTs”.

Line 191, the apical co-localization seems partial, and I recommend using a Pearson’s correlation.

Response: Thank reviewer’s suggestion. We select an ookinete apical area to calculate the Pearson correlation coefficient between APR2 and SPMT signals. The values of Pearson correlation coefficient were provided on the bottom of the revised Fig. 3a.

Line 236, These results do not prove strong binding behavior as stated, but co-localization. An immunoprecipitation or proximity ligation (PL) microscopy assay is required to conclude that they interact.

Response: we changed the “only GFP-tagged APR2-N displayed strong MT-binding behavior” to “only GFP-tagged APR2-N displayed a co-localization with MTs” (line 231-233).

Line 269, It would be clearer just to state MRC5 cells.

Response: we changed the “in certain transfected mammalian cells” to “in certain transfected MRC5 cells” (line 265).

Line 326, 329 need quantification like lines 318 and 320.

Response: we added the quantification information (line 322-324).

Line 328 suggests not indicates.

Response: we changed the “indicates” to “suggests” (line 325).

Line 345, sentence should state ‘after genetic disruption’ (not after loss).

Response: changed (line 342-243).

Line 348, signals? Do the authors mean EM density? This requires clarification and the data should be quantified from numerous observations.

Response: Yes, the signal mean the EM density. In these experiments, a number of ookinetes were analyzed for both WT and mutant parasites. We observed the difference (see the quantification of these results below, not included in our revised Figure).

Line 369, ‘severely destroyed’ is too strong a conclusion in my opinion. See the merge in Figure 6F apr2mutant, they are still there but fainter (as per the reduced signal area and volume).

Response: We deleted the “severely” in the revised manuscript (line 367).

Line 391, the terminology “almost destroyed” is not the best scientific nomenclature, can an alternative be used?

Response: We changed the “almost destroyed” to “severely destroyed” (line 388).

Line 410, the number of micronemes in the posterior versus anterior halves of the ookinetes should be quantified to support this claim.

Response: In WT ookinetes, the micronemes were mainly localized in the anterior and apical area. After APR2 disruption, the micronemes lost relatively anterior and apical localization and redistribute in cytosol randomly, instead of moving to the posterior. It is not appropriate to count the micronemes in the posterior versus anterior. To further strengthen our explanation, we provided more ookinetes images (not included in the revised manuscript) in the below.

Line 487, for completion, do the authors have data for ARA1 mutant phenotypes for ookinete formation, crescent shape, gliding speed and oocyst numbers?

Response: Thanks for reviewer's question about ARA1 function. We are currently performing experiments in an independent work to elucidate the precise role of ARA1 in APR and SPMT regulation. We hope to present the ARA1's story soon.

For all blots, please indicate if possible, how many ookinetes were loaded into each gel lane for reproducibility of Western blots (eg. Figure S2, etc).

Response: We have added the information for the amount of ookinetes used for immunoblot analysis in the revised figure legends.

In Figure 1, while it is shown that mature (presumably stage V) ookinetes are produced less, it is not clear at what stage (stage I, II, III or IV) the ookinetes arrest at. Can the authors quantify stages and show when the difference(s) occurs (eg Figure 1N)? See Figure 2A for example of the different stages.

Response: We added a description “Approximately two thirds of APR2-null ookinetes arrest at the early stages and the rest develop to ookinetes with less cell bending and defective gliding” in the revised legend of Fig. 1n.

Figure S1. In panel D, a zygote was shown for 17XNL but a retort for APR2 mutant. This makes the figure more confusing (I thought the micrographs were to follow the schematic above, but this is not correct, and it took some decoding). Please show similar representatives of the same stages in the micrographs. To assist with the confusion, could the schematic be placed into supplementary, or separated further from the microscopy? In panel E, it is best to ANOVA not t-test here and throughout the study for multiple comparisons.

Response: According to the reviewer’s suggestion, we updated the Supplementary Fig. 1d showing similar representative cell images for both WT and apr2 mutant parasites.

We agreed with the reviewer’s comment on statistical analysis in Supplementary Fig. 1e, and re-analyzed these data using ANOVA as suggested.

Figure 2. In panel B, the Brightfield channel is too dark. In panel C, please indicate which channel is HA and which is Myc.

Response: We have adjusted the brightfield channel in the revised Fig. 2b and added the information for the protein tags in the revised legend for Fig. 2c.

Figure S3A. It is not clear in the cross section which parts are SPMT until the very high zoom. Can this be made clearer with a different zoomed inset or color?

Response: We provided a new cross section image with higher resolution in the revised Supplementary Fig. 3a.

In Figure 4C, Did FL APR2(1-1394)-GFP also bind Tubulin by IP?

Response: We performed the experiments to test the interaction between GFP-tagged full-length APR2 (GFP alone as a control) and MTs in the HEK 293T cell using Co-IP and also detected the interaction between them. Below are the results (not included in the revised manuscript).

In Figure S4A, B the data are not proof of MT binding (line 249), but of co-localization. Would require an immunoprecipitation or PL microscopy assay to prove they interact.

Response: Thank reviewer for pointing it out. We changed the “MT-binding” to “MT-localization” in the revised manuscript (Line 245).

In Figure 5B, the APR2-dN ookinete looked straighter than the apr2::gfp control

ookinete. Was this measured? It likely indicates the region of APR2 that contributes to the phenotype. Can this be quantified from other images and included?

Response: The reviewer noticed that the *APR2-ΔN* mutant ookinetes looked straighter (less bending) compared with WT. Yes, they were. In the beginning of the story (Fig. 1k), we presented these ookinetes morphology defects with quantification for the APR2-null ookinetes.

Figure 6, panels C and D require arrows to show the details that are altered in dAPR2.

Response: Thanks for reviewer's suggestion. We have added the blue arrows to mark the apical defects in the mutant parasite in the revised Fig. 6b, c, d and Figure legend.

Figure 7G, use of densitometry for the Western blot would be useful. How many times was this repeated? It would be helpful to briefly define the treatments in the figure legend.

Response: We have added the blot densitometry values in the revised Fig. 7g. Shown were representative results from two replicate experiments. A brief description of those treatments with different detergents was added in the revised figure legend, and more details could be seen in the Methods section.

Figure S7C, the data circles in the black histogram bars are camouflaged, can these be another color to see them?

Response: Thanks for the reviewer's suggestion. We have changed the color for the data circles in the Supplementary Fig. 7c.

Figure S8G, can the colloidal gold particles be quantification be provided to support the claim (line 515)?

Response: We added the information, including the average number for colloidal gold particles per ookinete and the number of ookinetes analyzed, in the revised Supplementary Fig. 8g (see below).

Reviewer #3 (Remarks to the Author):

This is an elegant study of the composition, disposition and function of the apical polar ring (APR) of *Plasmodium* ookinetes. The establishment of a motile and invasive polarized cell is a critical event in zygote to ookinete differentiation for this group of parasites to allow transmission through the mosquito vector, as well as being essential for other invasive stages of the life cycle.

This work uses genetic modification, ultrastructural studies and proteomics to make a considerable contribution to our understanding of APR structure.

The work has been cleanly executed, the data are clear and very nicely presented, and the results represent a significant contribution to the area.

I have little to criticize in the study and believe it will be of considerable interest to, for example, cell biologists, particularly those with an interest in apicomplexan parasites.

Response: We thank reviewer for the positive comments on our manuscript.

Minor points:

Line 51; my understanding is that the pellicle includes the parasite plasma membrane.

Response: We agree with reviewer and have accordingly modified the text (Line 48).

Throughout the text and supplementary figures (including figure axes); the English needs to be checked for typos and grammar and corrected as appropriate.

Response: We have re-checked the typos and grammar throughout texts and figures.

Reviewer #4 (Remarks to the Author):

The study by Qian et al. characterizes a defining structure of the apicomplexan phylum, the apical polar ring (APR), with unprecedented detail in the ookinete. They provide an extensive phenotypic analysis of a critical component of this structure, APR2, and expand the APR proteome by some novel component. This study presents an excellent body of work including very attractive super-resolution imaging that unpacks the role of APR2 in a series of logical steps. Their conclusions are supported by the remarkably large and well-controlled dataset, which is even complemented by analyses in mammalian cell model, and only requires minor improvements. The study provides very new insights into the organization critical for multiple life cycle stages of the parasite with relevance for other apicomplexan species. Taken together the authors provide an excellent cell biological framework for any further study of the APR as an atypical microtubule organizing center. Hence, I have no reservations to recommend this manuscript for publication in your journal after the following comments have been properly addressed.

Response: We thank reviewer for the positive comments on our manuscript.

Major comments

Fig. 1A: The integration PCR results for the *apr2* mutant strains, which is also mentioned in methods, should be shown (even though the phenotypes look convincing).

Response: The PCR genotyping results for *apr2* and other mutant strains were prepared but not included in the initial submission. In the revised manuscript, we have added a supplementary file (Supplementary Fig. 11) which provided the PCR genotyping results for all the modified parasite strains generated in this study.

Fig. 3B: Negative controls with only one antibody for the PLA in need to be shown (as well as for Fig. 9B).

Response: We thanks for reviewer's suggestion on the PLA assay. Indeed, negative controls with only one antibody for PLA were also performed in parallel. Expectedly these negative controls with one antibody did not produce detectable PLA signal. Below are the results (not included in the revised manuscript). Because the space limit, these

results for one antibody control group were not included in the Fig. 3b.

In this study, the wildtype parasite (17XNL) with two antibodies (anti-HA and anti-Tubulin) were used as negative control for PLA in the Fig. 3b. To our knowledge, this format for PLA negative controls was also used in the publications.

Fig. 4E: Analyses of changes of microtubule dynamics shown in Fig. 4E are very enlightening. Did something prevent the orders from using the full length APR2 (seen in Fig. 4A) in this experiment? If possible, it would strengthen the conclusions by the authors about the function of APR2.

Response: We agree with reviewer's comments. We did plan the *in vitro* MT-binding assay for the full-length APR2 protein, but unfortunately failed to obtain the purified recombinant full-length APR2-GFP protein from mammalian cells after several trials. Fortunately, successful purifications of three overlapped fragments covering the full length of APR2 provided an alternative in this study.

If possible, please show colocalization analysis of ARA1 with APR2 to learn more about their relative positioning.

Response: The co-localization results for APR2 and ARA1 were presented in a double-tagged strain (GFP for APR2 and HA for ARA1) in the Fig. 8h (see below).

Fig. 7B: Add total tubulin intensity to highlight microtubule stability difference.

Response: According to reviewer's suggestion, we analyzed the MT tracker (SiR-Tubulin) signal in the ookinetes in the Fig.7a and observed the decreased MT signal in the APR2-null parasites as expected (see below, not included in the revised manuscript).

Fig. S7: Add IFA images (ideally ExM) of micronemes in *apr2* KO to confirm EM images and claims about microneme phenotype.

Response: We thank reviewer for suggesting additional image (U-ExM) analysis about microneme phenotype. Although successful adaptation of U-ExM in apicomplexan parasites, this method was used for only MTs or cytoskeleton protein analysis in the parasites considering its technical features. To our knowledge, utilization of U-ExM for membrane organelles analysis is not reported yet.

To further address the reviewer's concern, we provided some more TEM images (see below) for WT and *apr2* KO parasites showing the localization of micronemes in the ookinetes. The area with micronemes were labeled with white dash line

17XNL (wildtype)

$\Delta apr2$

Fig. 9A: The panel supposedly showing the Y2H results for interaction with APR2-F1 fragment seems completely dark even though at least some background should be visible. Please add panel or adjust contrast.

Response: We have adjusted the image contrast for Fig. 9a as reviewer suggested.

Minor comments

Line 17: Find another formulation than “possessing”.

Response: we changed the “possessing” to “containing” (Line 17).

Line 188: complete title.

Response: we changed the subtitle “APR2 associates with SPMTs at apical” to “APR2 associates with apical SPMTs” (Line 184).

Line 216: Replace the term “fluidity” with an alternative, like turnover.

Response: we have changed “fluidity” to “turnover” (Line 211).

The authors could comment a bit more about the inner and outer APR highlighted by Ferreira et al. in the context of the likely relative position of APR2 and associated proteins.

Response: in preparing the manuscript, we noticed that Ferreira *et al.* performed a fine structure analysis for SPMT and APR at three zoite stages in *Plasmodium*. We have cited this publication and commented their work in the discussion section. Using different methods, we and Ferreira uncovered a similar APR 3D-structures in the ookinetes. They also identified two layers (inner and outer) of APR structure in native cellular context. Because of technical limitation, currently we could not assign the APR proteins into APR inner and outer layers or other subcompartments. Precise localization of these proteins within APR need more works in the future.

REVIEWERS' COMMENTS

Reviewer #1 (Remarks to the Author):

The authors have thoroughly addressed my comments. I do not have further comments. I read through the responses to comments from other reviewers. The authors did a great job, and the updated manuscript is improved and well-described.

Reviewer #2 (Remarks to the Author):

The authors have addressed my concerns. I strongly suggest that the authors consider including the new data provided in rebuttal that is "(not included in the revised manuscript)" in the supplementary, as it will be helpful for the readership.

Reviewer #4 (Remarks to the Author):

All comments raised during my first review have been sufficiently addressed in the revised version. I have no further criticism or suggestions concerning the presented manuscript and want to congratulate the authors on their highly insightful work.

Response to Reviewer Comments on the manuscript [NCOMMS-22-27179A]:

Reviewer #1 (Remarks to the Author):

The authors have thoroughly addressed my comments. I do not have further comments. I read through the responses to comments from other reviewers. The authors did a great job, and the updated manuscript is improved and well-described.

Response: We thank reviewer for the positive comments on our manuscript.

Reviewer #2 (Remarks to the Author):

The authors have addressed my concerns.

I strongly suggest that the authors consider including the new data provided in rebuttal that is "(not included in the revised manuscript)" in the supplementary, as it will be helpful for the readership.

Response: We thank reviewer for the positive comments on our manuscript. There are already 15 supplementary figures, tables, and datasets in the manuscript. We are going to leave the new data in the rebuttal, which is accompanied with the manuscript as required by the journal and is open to the readers.

Reviewer #4 (Remarks to the Author):

All comments raised during my first review have been sufficiently addressed in the revised version. I have no further criticism or suggestions concerning the presented manuscript and want to congratulate the authors on their highly insightful work.

Response: We thank reviewer for the positive comments on our manuscript.